# iMyoblasts for ex vivo and in vivo investigations of human myogenesis and disease modeling

**Dongsheng Guo[1,2†], Katelyn Daman[1,2†], Jennifer JC Chen[1], Meng-Jiao Shi[1], Jing Yan[1], Zdenka Matijasevic[1,3], Amanda M Rickard[4], Monica H Bennett[4], Alex Kiselyov[4], Haowen Zhou[5], Anne G Bang[5], Kathryn R Wagner[6], René Maehr[7], Oliver D King[1], Lawrence J Hayward[1,2], Charles P Emerson Jr[1,2]\***

[1]Wellstone Muscular Dystrophy Program, Department of Neurology, University of Massachusetts Chan Medical School, Worcester, United States; [2]Li Weibo Institute for Rare Disease Research, University of Massachusetts Chan Medical School, Worcester, United States; [3]Transgenic Animal Modeling Core, University of Massachusetts Chan Medical School, Worcester, United States; [4]Genea Biocells, La Jolla, United States; [5]Conrad Prebys Center for Chemical Genomics, Sanford Burnham Prebys Medical Discovery Institute, La Jolla, United States; [6]Center for Genetic Muscle Disorders, Kennedy Krieger Institute, Baltimore, United States; [7]Program in Molecular Medicine, University of Massachusetts Chan Medical School, Worcester, United States

**\*For correspondence:** charles.emersonjr@umassmed.edu

[†]These authors contributed equally to this work

**Abstract** Skeletal muscle myoblasts (iMyoblasts) were generated from human induced pluripotent stem cells (iPSCs) using an efficient and reliable transgene-free induction and stem cell selection protocol. Immunofluorescence, flow cytometry, qPCR, digital RNA expression profiling, and scRNA-Seq studies identify iMyoblasts as a *PAX3+/MYOD1+* skeletal myogenic lineage with a fetal-like transcriptome signature, distinct from adult muscle biopsy myoblasts (bMyoblasts) and iPSC-induced muscle progenitors. iMyoblasts can be stably propagated for >12 passages or 30 population doublings while retaining their dual commitment for myotube differentiation and regeneration of reserve cells. iMyoblasts also efficiently xenoengrafted into irradiated and injured mouse muscle where they undergo differentiation and fetal-adult MYH isoform switching, demonstrating their regulatory plasticity for adult muscle maturation in response to signals in the host muscle. Xenograft muscle retains PAX3+ muscle progenitors and can regenerate human muscle in response to secondary injury. As models of disease, iMyoblasts from individuals with Facioscapulohumeral Muscular Dystrophy revealed a previously unknown epigenetic regulatory mechanism controlling developmental expression of the pathological *DUX4* gene. iMyoblasts from Limb-Girdle Muscular Dystrophy R7 and R9 and Walker Warburg Syndrome patients modeled their molecular disease pathologies and were responsive to small molecule and gene editing therapeutics. These findings establish the utility of iMyoblasts for ex vivo and in vivo investigations of human myogenesis and disease pathogenesis and for the development of muscle stem cell therapeutics.

## Editor's evaluation

This is an interesting and systematically constructed paper developing an iPSC–myoblast platform. It covers the generation of the cell system, and its detailed description. Assessments are made as to the extent this iPSC system recapitulates Limb Girdle Muscular Dystrophy and Walker Warburg Syndrome. The findings promote the view that iPSC myoblasts have a potential in studying muscle stem cell functions and for therapeutic development.

## Introduction

The technologies for reprogramming human somatic cells into induced pluripotent stem cells (iPSCs) (*Takahashi et al., 2007*) and for inducing specific differentiated cell types are providing extraordinary opportunities for investigating mechanisms of human tissue differentiation, the molecular pathology of diseases, and therapeutic development. Much of the research on iPS cell-type induction has focused on optimizing the production of differentiated cells to provide a platform for investigations of disease pathologies (*Ardhanareeswaran et al., 2017*; *Hashimoto et al., 2016*; *Georgomanoli and Papapetrou, 2019*; *Heslop and Duncan, 2019*; *van Mil et al., 2018*). Less attention has been given to generation of lineage-specific human stem cells and progenitors to enable studies of tissue and organ development, genetic and epigenetic disease mechanisms, and stem cell therapeutics.

The goal of our study has been to isolate and propagate myogenic stem cells from human iPSC cultures in response to gene-free myogenic induction and cell growth selection and to establish the utility of these myoblast stem cells for molecular investigations of human myogenesis and muscular dystrophies. Here, we report an efficient and reliable transgene-free myogenesis protocol to generate human skeletal muscle stem cells (iMyoblasts) from healthy control (Ctrl) and patient iPSCs. This protocol efficiently produces a stably committed and expandable population of PAX3+/MYOD1+ iMyoblasts that differentiate as regenerative stem cells ex vivo in cell culture and in vivo in muscle xenografts in irradiated and injured mouse tibialis anterior (TA) muscle, which maintain a renewable PAX3+ iMyoblast population. iMyoblast muscle xenografts undergo fetal-to-adult MYH isoform switching demonstrating their plasticity to respond to maturation signals provided by the host adult muscle. Finally, we show that iMyoblasts generated from Facioscapulohumeral Muscular Dystrophy (FSHD) Type 1 (FSHD1), Limb-Girdle Muscular Dystrophy (LGMD) R7 and R9 (formerly LGMD2G and 2I), and Walker Warburg Syndrome (WWS) patient iPSCs model the molecular pathologies of these diseases.

## Results

### Isolation of iMyoblasts by iPSC transgene-free induction and reserve cell selection

We developed a two-step protocol to isolate iMyoblasts from cultures of Ctrl and patient iPSCs, using transgene-free iPSC myogenic induction in combination with reserve cell selection (*Figure 1A*). Human iPSC lines for these studies were generated by reprogramming bMyoblasts and fibroblasts isolated from muscle biopsies of adult FSHD1 and Ctrl subjects (*Homma et al., 2012*; *Jones et al., 2012*), or dermal fibroblasts from subjects with early onset FSHD1, LGMDR7, LGMDR9, and WWS (*Kava et al., 2013*). The first step of the iMyoblast protocol was transgene-free iPSC myogenesis induction using commercially available reagents (*Caron et al., 2016*; Amsbio, Skeletal Muscle differentiation Kit) (*Figure 1—figure supplement 1A*). This three-stage iPSC myogenesis protocol induces cultures of Ctrl and disease iPSCs to sequentially upregulate expression of muscle master regulators, *PAX3* (S1 stage) and *MYOD1* (S2 stage), and the muscle differentiation marker *MYH8* (S3 Stage) (*Caron et al., 2016*), as assayed by qPCR (*Figure 1—figure supplement 1B*). Gene expression was also assayed in the FSHD1 and Ctrl Embryonic Stem Cell (ESC) lines originally used to develop and optimize this induction protocol to assure that iPSCs and ESCs respond similarly to this transgene-free myogenesis induction protocol (*Figure 1—figure supplement 1B*). These studies established that Ctrl and disease iPSC and ESC lines robustly upregulated expression of *PAX3*, *MYOD1*, and *MYH8* on the order of 1000-fold during S1, S2, and S3 stages of myogenic induction, validating the consistency and efficiency of the transgene-free induction protocol. Immunofluorescence (IF) assays showed that 90% of cells in S2 stage cultures were MYOD1+, and 80% of cells in S3 stage cultures were MF20+ and predominantly mononucleated iMyocytes (*Figure 1A*), similar to the first myogenic cells to differentiate in the embryo (*Lee et al., 2013*). *PAX7* expression was detected at 100-fold lower levels than *PAX3* during S1 induction (*Figure 1—figure supplement 1B*), consistent with earlier findings that PAX7+ cells are a minor cell population induced by transgene-free myogenesis protocols (*Chal et al., 2015*; *van der Wal et al., 2018*).

The second step – for isolation of iMyoblasts – utilized reserve cell selection, as adapted from a protocol previously employed to isolate quiescent myogenic cells generated during differentiation of C2C12 myotube cultures by growth factor stimulation (*Yoshida et al., 1998*; *Laumonier et al., 2017*).

**eLife digest** Muscular dystrophies are a group of inherited genetic diseases characterised by progressive muscle weakness. They lead to disability or even death, and no cure exists against these conditions.

Advances in genome sequencing have identified many mutations that underly muscular dystrophies, opening the door to new therapies that could repair incorrect genes or rebuild damaged muscles. However, testing these ideas requires better ways to recreate human muscular dystrophy in the laboratory.

One strategy for modelling muscular dystrophy involves coaxing skin or other cells from an individual into becoming 'induced pluripotent stem cells'; these can then mature to form almost any adult cell in the body, including muscles. However, this approach does not usually create myoblasts, the 'precursor' cells that specifically mature into muscle during development. This limits investigations into how disease-causing mutations impact muscle formation early on.

As a response, Guo et al. developed a two-step protocol of muscle maturation followed by stem cell growth selection to isolate and grow 'induced myoblasts' from induced pluripotent stem cells taken from healthy volunteers and muscular dystrophy patients. These induced myoblasts can both make more of themselves and become muscle, allowing Guo et al. to model three different types of muscular dystrophy. These myoblasts also behave as stem cells when grafted inside adult mouse muscles: some formed human muscle tissue while others remained as precursor cells, which could then respond to muscle injury and start repair.

The induced myoblasts developed by Guo et al. will enable scientists to investigate the impacts of different mutations on muscle tissue and to better test treatments. They could also be used as part of regenerative medicine therapies, to restore muscle cells in patients.

iMyoblast reserve cells were recovered by activation of proliferation of undifferentiated cells resident in differentiated S3 muscle cultures using the same growth-factor-rich medium used to maintain proliferative cultures of adult biopsy myoblasts (bMyoblasts). This myoblast growth medium promotes the efficient recovery of a proliferative, myogenic cell population of MYOD1+ cells, referred to as iMyoblasts (*Figure 1A*). The iMyoblast protocol has been successfully applied to the isolation of iMyoblast lines from Ctrl iPSCs as well as classic and early onset FSHD1, WWS, LGMDR7, and LGMDR9 iPSCs.

To validate the iMyoblast technology and establish its utility for disease studies, subsequent experiments were conducted in parallel with FSHD1 and Ctrl iMyoblasts, FSHD1 and Ctrl bMyoblasts, and also WWS, LGMDR7, and R9 iMyoblasts. In growth factor-rich medium, bMyoblasts and iMyoblasts expressed muscle master regulatory genes *PAX3* and *MYOD1* whereas adult bMyoblasts expressed *PAX7* in addition to *PAX3* and *MYOD1* (*Figure 1B*). iMyoblasts proliferated in myoblast growth medium with 12 hr cell doubling times and could be expanded as primary lines for more than 12 passages (>30 population doublings) while cell-autonomously maintaining expression of *PAX3* and *MYOD1* and the commitment to differentiate in response to growth factor free medium, as assayed by expression of *MYH8* and *CKM* (*Figure 1C*). During their differentiation, iMyoblasts fused to form multinucleated iMyotubes (*Figure 1A*). Proliferating FSHD1 and Ctrl iMyoblasts expressed cell surface markers typical of fetal and adult myogenic cells, including CD82 (*Alexander et al., 2016*; *Pakula et al., 2019*) as well as CD56, CD318 (*Uezumi et al., 2016*), ERBB3, and NGFR (*Hicks et al., 2018*; *Figure 1D*). FAC-sorted CD82+/CD56+ and CD82+/CD56− iMyoblasts both differentiated and fused to form MF20+ iMyotubes, validating the commitment of CD82+ iMyoblasts to differentiate (*Figure 1E*). Finally, iMyoblasts retained their dual commitment to both differentiate and generate reserve cells that could be recovered by growth medium stimulation as MYOD1+ iMyoblasts (tMyoblasts) that can fuse and differentiate as MF20+ and MEF2C + iMyotubes (*Figure 1—figure supplement 2*).

## scRNA-Seq identifies iMyoblasts as a myogenic cell lineage

Single-cell RNA sequencing (scRNA-Seq) was used to define the iMyoblast transcriptome and compare it to transcriptome signatures of adult muscle bMyoblasts, and S1 and S2 stage cells undergoing iPSC myogenic induction. For these studies, iMyoblasts were generated by iPSC induction and reserve cell propagation from S3 cultures. iPSCs were reprogrammed from parental CD56+ muscle biopsy cells

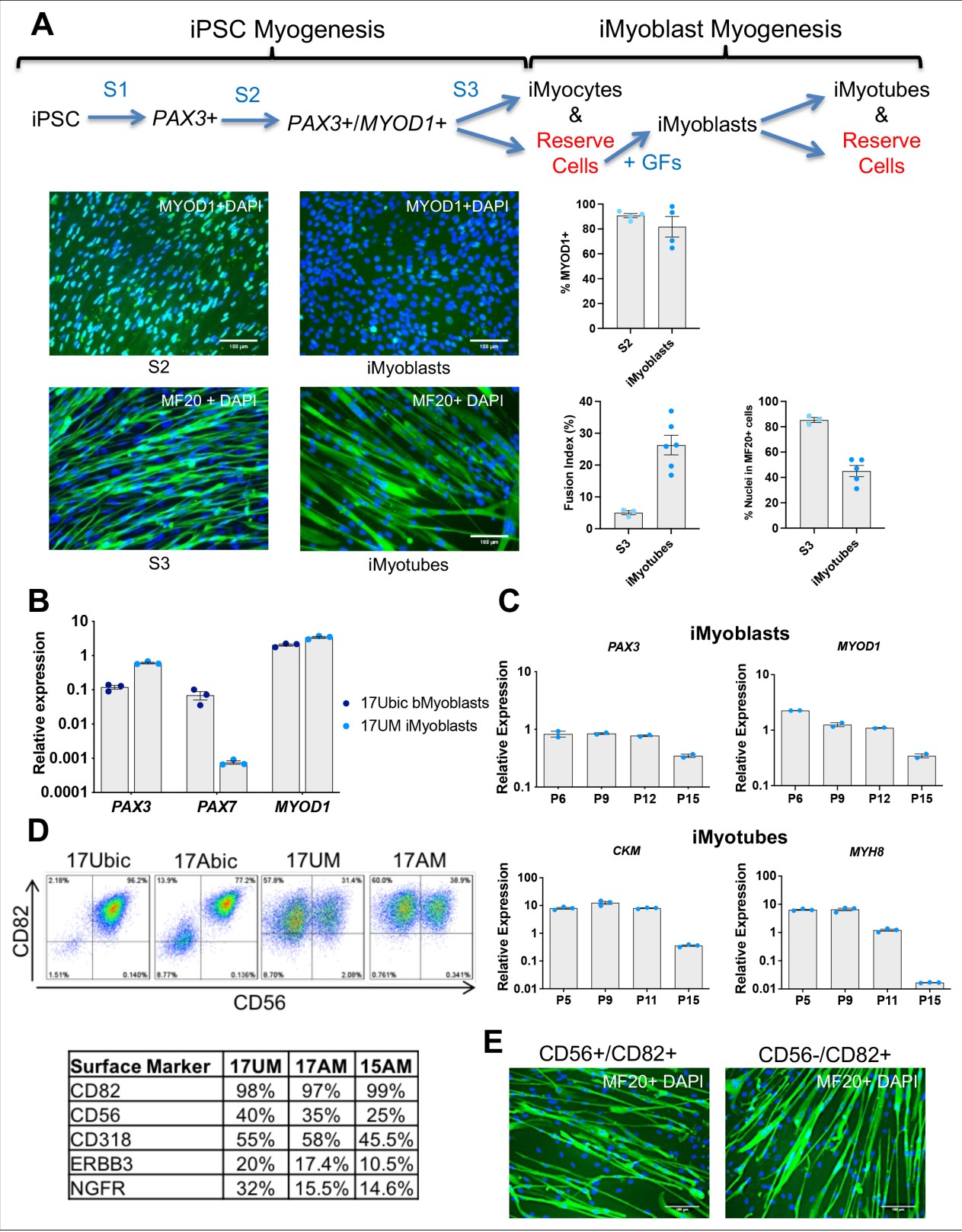

**Figure 1.** Isolation and characterization of iMyoblasts. (**A**) Schematic of a three-stage transgene-free iPSC induction, iMyoblast reserve cell isolation, and iMyotube differentiation protocol. Images of S2 cells and iMyoblasts immunostained with MYOD1 antibody, and S3 iMyocytes and iMyotubes immunostained with MF20 myosin antibody. Nuclei are stained with DAPI. Scale bars=100 μm. Quantification of % MYOD1+ S2 cells and iMyoblasts, fusion index (the percentage of nuclei within MF20+ cells containing ≥2 nuclei ) and % nuclei within MF20+ cells for S3 cells and iMyotubes are shown

*Figure 1 continued on next page*

*Figure 1 continued*

on the right. For quantification, each dot corresponds to the % MYOD1+ cells or fusion index in an individual image. Data are presented as mean ± SEM for each condition. (**B**) qPCR assays of *PAX3*, *PAX7*, and *MYOD1* in bMyoblasts (17Ubic) and iMyoblasts (17UM) normalized to *RPL13A*. (**C**) qPCR assays normalized to *RPL13A* of proliferating (top, iMyoblasts) or Day 7 differentiated (bottom, iMyotubes) Ctrl 17UM iMyotubes with increasing passage (P) numbers. (**D**) Flow cytometry of CD56 and CD82 cell surface markers for bMyoblasts (17Ubic, 17Abic) and iMyoblasts (17UM, 17AM). Table below summarizes flow cytometry assays of iMyoblast surface markers in Ctrl (17UM) and FSHD1 (17AM, 15AM) cell lines. (**E**) MF20 immunostaining of CD56+/CD82+ or CD56-/CD82+ Ctrl (17UM) iMyotubes after 7 days of differentiation. Scale bars=100 µm.

The online version of this article includes the following source data and figure supplement(s) for figure 1:

**Source data 1.** Source data for *Figure 1*.

**Figure supplement 1.** Transgene-free myogenic induction of FSHD1 and Ctrl iPSC and ESC and FKRP LGMDR9 and LGMDR7 iPSCs.

**Figure supplement 1—source data 1.** Source data for *Figure 1—figure supplement 1*.

**Figure supplement 2.** Reserve cell isolation of induced tertiary Myoblasts.

**Figure supplement 2—source data 1.** Source data for *Figure 1—figure supplement 2*.

of six subjects, including two classic FSHD1 subjects (15A and 30A), one early onset FSHD1 subject (17A), and their unaffected Ctrl family members (15V, 17U, and 30W). These same six subjects were the source of S1 and S2 stage cells derived from iPSCs and unsorted adult muscle biopsy cells. Normalization of Unique Molecular Identifier (UMI) counts, cell-cycle estimation, dimension-reduction, and cell cluster identification were performed using Seurat (*Figure 2A*). The Uniform Manifold Approximation and Projection (UMAP) plots grouped Ctrl and FSHD1 cells together in each of the five main clusters (*Figure 2B*), as expected as the FSHD1 disease genes are expressed in differentiated myotubes and not in proliferating progenitors. Transcriptomes of differentiated iMyotubes, bMyotubes, and S3 stage muscle were not investigated in this study.

UMAP grouped Ctrl and FSHD iMyoblasts into clusters that were distinct from S1, S2, bMyoblast (bMyo), and biopsy mesodermal cell (bMes) clusters. Each cell type cluster included subsets of cells expressing genes of the different stages of the cell cycle and cells contributed by all six subjects, consistent with a reliable and sensitive UMAP segregation (*Figure 2B*). Each of the five clusters had a distinct gene expression signature of myogenic regulatory genes, differentiation genes, and cluster marker genes that further validated their myogenic identities (*Figure 2C*). The iMyoblast cluster expressed *PAX3* and *MYOD1*, the bMyoblast cluster expressed *PAX3* and *MYOD1* as well as *PAX7* and *MYF5*, and the bMes non-myogenic cluster expressed *PDGFRA*, a mesodermal cell marker (*Evseenko et al., 2010*; *Joe et al., 2010*; *Uezumi et al., 2010*; *Ding et al., 2013*). bMyo and bMes cells expressed *NFIX*, a regulator of the switch from embryonic to fetal myogenesis (*Messina et al., 2010*), also expressed by iMyoblasts at lower levels (*Supplementary file 1*). Cells in the S1 cluster expressed *PAX3* at higher levels than cells in the iMyoblast cluster and expressed *LIN28A*, which encodes an RNA binding protein controlling self-renewal and differentiation (*Shyh-Chang and Daley, 2013*). S2 cells had heterogeneous morphology (*Figure 2—figure supplement 1*) and could be subdivided into a proximal S2A subcluster expressing *PAX3* and a more distal S2B subcluster expressing *MYOD1*, *MYOG*, and *MYH8* muscle differentiation genes, and $G_1$ cell cycle markers, consistent with their developmental progression from pre- to post-differentiation stages. These findings establish iMyoblasts as a myogenic cell with a transcriptome distinct from bMyoblasts and S1 and S2 stage myogenic cells.

## Pathway analysis of the iMyoblast transcriptome

Differences in gene expression between iMyoblasts, bMyoblasts, bMes, S1, S2A, and S2B cell classes were quantitated and subjected to pathway analysis using edgeR (*Robinson et al., 2010*). This analysis was based on pseudo-bulk expression profiles (*Tung et al., 2017*) derived from scRNA-Seq data by summing counts of all cells from the same cell class and donor to avoid spurious results due to pseudoreplication (*Hurlbert, 1984*). The 15 pairwise comparisons among the six cell classes each identified at least 4600 differentially expressed coding and non-coding genes at a false discovery rate (FDR)<0.05 (*Supplementary file 1*). It is possible that some of these may be attributable to technical batch effects, as cells of the same type (i.e., S1, S2, iMyoblast, or bMyoblast) from all donors were multiplexed during the single-cell encapsulation and library construction. This caution does not apply to comparisons of S2A versus S2B and bMyo versus bMes, although these may be biased toward

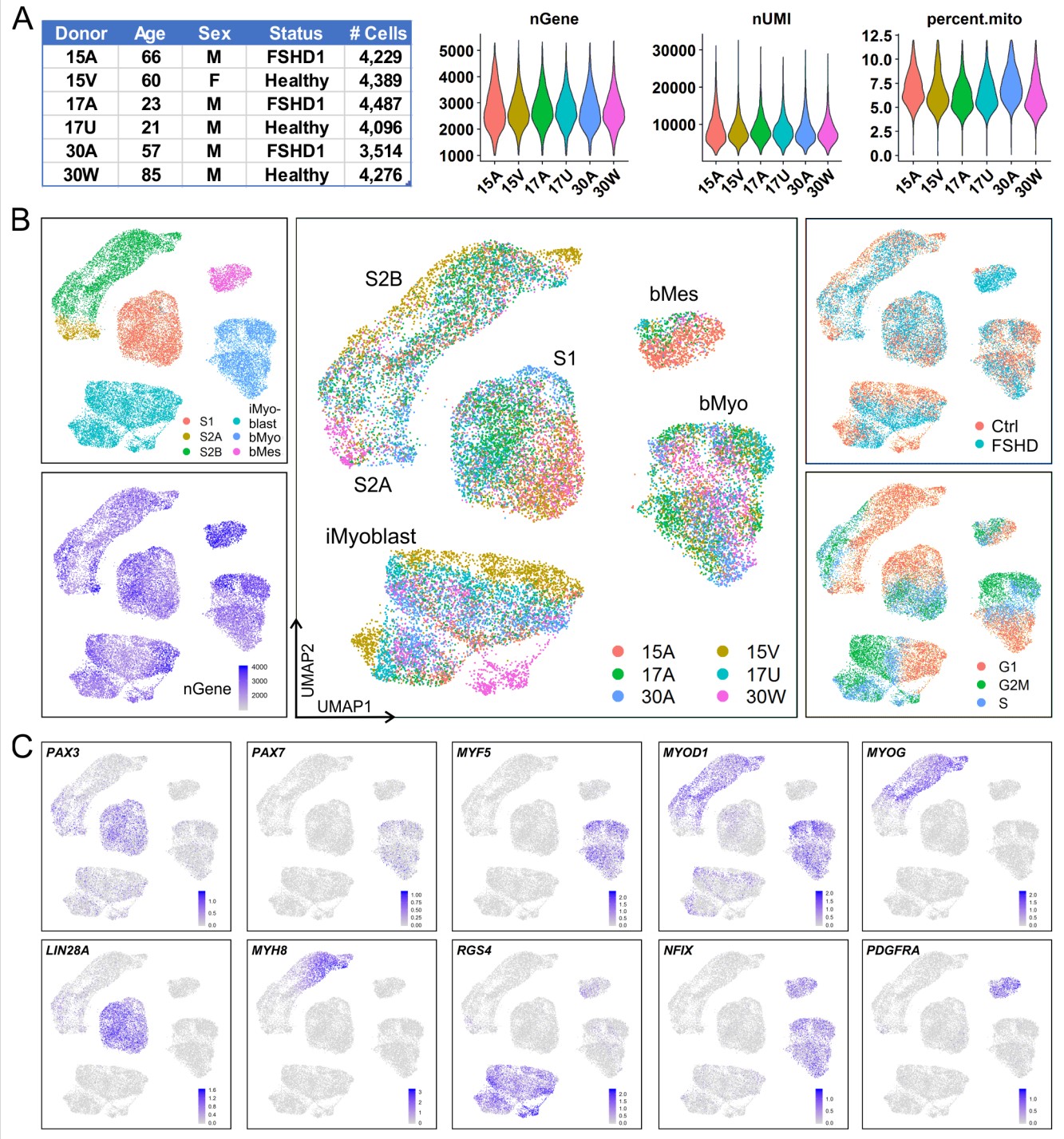

**Figure 2.** iMyoblasts have a distinct gene expression signature compared to S1, S2, and muscle biopsy cells. (**A**) Single-cell transcriptome sequencing (scRNA-Seq) was performed on S1, S2, iMyoblasts and bMyoblasts from three FSHD1 and three healthy (Ctrl) donors. A total of 24,991 cells satisfied criteria that included 1000–5500 genes detected per cell (nGene), <40,000 UMI detected per cell (nUMI), and <12% of total reads from mitochondrial genes. (**B**) iPSC-induced S1 and S2 cultures, iMyoblasts and primary biopsy cultures segregated into distinct clusters in UMAP plot (upper left panel). S2 cells were segregated into S2A and S2B subclusters. Biopsy cells were segregated into a myogenic cell cluster (bMyo) and a non-myogenic mesodermal cell cluster (bMes). UMAP plots with color-scales indicating nGene, donor identities, disease group, and estimation of cell cycle state for each cell are also shown. (**C**) Expression of representative myogenesis and cluster marker genes in UMAP plots.

The online version of this article includes the following figure supplement(s) for figure 2:

**Figure supplement 1.** Morphology of different cell classes used in scRNA-Seq.

small p-values since these clusters are defined based on the same transcriptomic data that is being compared between clusters (*Zhang et al., 2019*). Tests of differential expression between FSHD1 and Ctrl samples from the same cell cluster are not subject to the biases above, but their power is limited by the small number of subjects and by low expression of *DUX4* and its targets, as expected for proliferating FSHD1 cells. For this reason, we did not focus on these comparisons, though as a caution we note that the single gene, *AC004556.1*, that had FDR<0.05 in these comparisons appears to be an annotation artifact: a common variant in the gene *MRPL23* that happens to occur in these FSHD1 subjects but not the Ctrl subjects caused reads from *MRPL23* to be assigned to the gene *AC004556.1* on an unlocalized scaffold instead.

A gene expression dot plot was generated from scRNA-Seq data to illustrate graphically the quantitative differences in gene expression and cell expression frequency across the six cell classes for each of the six subjects (*Figure 3*). This analysis focused on a manually curated set of differentially-expressed genes chosen based on their known developmental and regulatory functions in myogenesis. Some of these curated genes showed cell cluster-specific expression but many were expressed by multiple cell clusters, likely reflecting their shared developmental histories and myogenic functions. However, this dot plot illustrates that each cluster population has a distinct gene expression profile shared by cells from all six subjects in each cluster. These data further establish that iMyoblasts have a myogenic transcriptome that includes extracellular matrix (ECM) components, signaling molecules, and transcriptional regulators distinct from bMyoblasts and iPSC-induced S1 and S2 stage myogenic cells.

Pathway analysis was also used to compare the transcriptomes of iMyoblasts with myogenic cells in other cell classes based on biological function. Gene ontology (GO) and KEGG pathway enrichment analyses were performed using the edgeR functions goana and kegga (*Young et al., 2010*), applying more stringent cutoffs on differential expression, p-value<1E−06 and |log2(FC)|>1, and with enrichment analyses performed separately for upregulated and downregulated genes. The top-ranked GO and KEGG categories for each of the 15 pairwise comparisons, sorted by p-value for enrichment, are listed in *Supplementary file 2*, as summarized below.

The top-ranked categories for the iMyoblast versus bMyoblast comparison included categories of known myogenic genes, including ECM, focal adhesion, and migration/chemotaxis (*Gillies and Lieber, 2011*; *Csapo et al., 2020*; *Thorsteinsdóttir et al., 2011*; *Rayagiri et al., 2018*), signaling (*Chal and Pourquié, 2017*), and transcription (*Berkes and Tapscott, 2005*; *Buckingham and Relaix, 2015*). Both the upregulated and downregulated genes were significantly enriched for ECM genes, including distinct collagen gene isoforms, with *COL4A1, COL4A2, COL4A5, COL4A6, COL8A1, COL11A1*, and *COL13A1* UP in iMyoblasts compared to bMyoblasts, and *COL6A1, COL6A2, COL6A3, COL1A2, COL5A2, COL7A1, COL8A2,* and *COL22A*1 UP in bMyoblasts compared to iMyoblasts. Other ECM genes UP in iMyoblasts included *AGRN, QSOX1, DSP, ECM1, SLIT2, EXT1, EXTL1*, and *EXTL3*, and those UP in bMyoblasts included *DRAXIN, NFASC, EVL,* and *ELN*. iMyoblasts and bMyoblasts also differentially expressed members of matrix processing enzyme gene families, including *MMP, ADAMTS,* and *ADAM,* and members of matrix regulatory protein gene families *ITG, KRT, CDH, SEMA,* and *LAM*, all of which have established regulatory functions during embryonic and adult myogenesis.

Signaling was among the top-ranked GO and KEGG categories and included receptor regulatory activity, receptor ligand activity, and signaling receptor activator activity (enriched among genes UP in bMyoblasts compared to iMyoblasts), and PI3K-Akt, MAPK, Rap1, and Ras pathways (enriched among genes UP in iMyoblasts vs. bMyoblasts). These pathways include differentially expressed *FGF, WNT,* and *FZD* gene family members. Additionally, *TGFB, PDGFA, EPHA2* and *EPHB2, NOG, IGF2BP1* and *IGF2BP3,* and *HMGA2* were UP in iMyoblasts, while *GDNF, VEGFA, BMP4, BMP7, WISP1, SULF1,* and *GREM2* were UP in bMyoblasts.

Transcription categories included DNA-binding transcription activators, for which specific genes UP in bMyoblasts included *KLF4, SOX8, SIX2, PITX3, SCX, SNAI1, SNAI2,* and *SMAD1,* and genes UP in iMyoblasts included *GATA3, GATA6, HAND2, MEIS2, GLI2, NOTCH1, ETS1,* and *ETS2*. The top-ranked KEGG categories for genes UP in bMyoblasts compared to iMyoblasts included mineral absorption, complement and coagulation cascades, arachidonic acid metabolism, and retinoic acid metabolism, whose functions in adult myogenesis are currently unknown.

The results above focus on differences between iMyoblasts and bMyoblasts, but similarities between these cell types can be seen by contrasting each with clusters S1 and S2A, cells in earlier

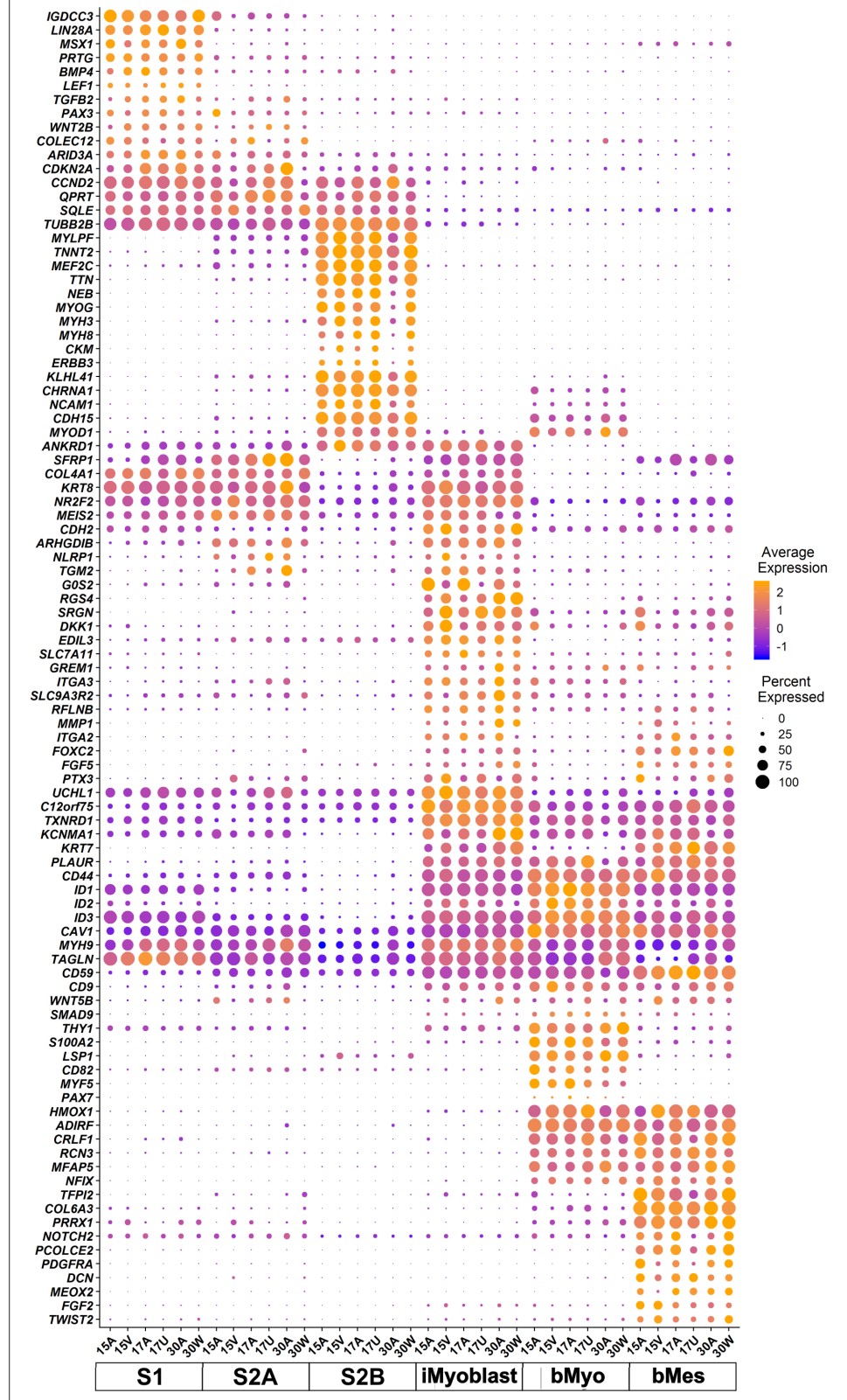

**Figure 3.** scRNA-seq transcriptome signatures of iMyoblasts from FSHD1 and Ctrl subjects compared to S1, S2A, S2B, bMyo, and bMes myogenic cell classes. Dot plot of manually curated genes in cells from six different subjects for each of the six cell classes (bottom row). Purple to orange colors define the low to high average expression of each gene (vertical row) in cells of each subject in each of the six cell classes (columns), centered

*Figure 3 continued on next page*

*Figure 3 continued*

to have mean=0 and scaled to have SD=1; positive values indicate upregulation and negative values indicate downregulation compared to the average expression across all cell groups. Dot sizes indicate the fraction of cells expressing each gene.

stages of myogenic induction. In all four pairwise comparisons between earlier stages (S1 or S2A) and committed iMyoblasts or bMyoblasts stages, genes UP in the committed myogenic stages are enriched for positive regulation of cell migration and TGFβ signaling, while genes UP in the earlier stage are enriched in steroid and cholesterol biosynthesis categories, likely to inhibit replicative stress and replication check point activation leading to cell cycle arrest (*Replogle et al., 2020*).

These findings validate the identity of iMyoblasts as a bona fide *PAX3/MYOD1* myogenic cell, distinct both from S1 and S2 cells at early developmental stages of iPSC induction and from adult *PAX7/PAX3/MYF5/MYOD1* bMyoblasts.

## iMyoblast differentiation ex vivo

The differentiation of Ctrl, FSHD1, and LGMDR7 iMyoblasts was compared in cultures using growth factor-free N2 medium to induce myotube differentiation. iMyotube cultures from these iMyoblasts upregulated *CKM* and *MYH8* muscle genes with similar kinetics, as determined by qPCR assays (*Figure 4A*). iMyotubes expressed low levels of adult *MYH1* compared to bMyotubes (*Figure 4A*), consistent with their identity as a fetal/embryonic lineage. Ctrl and FSHD1 iMyotubes similarly upregulated the expression of muscle genes in both N2 and Opti-MEM growth factor-free media (*Figure 4— figure supplement 1A*). Expression levels of myogenic regulators *PAX3*, *MYOD1*, and *MYOG* and *CKM* and *MYH8* differentiation genes varied by cell line but were not specifically impacted by whether iMyoblasts were derived from iPSCs reprogrammed from fibroblasts or bMyoblast parental cells (*Figure 4—figure supplement 1B*).

The effects of myogenic signaling modulators on iMyotube and bMyotube differentiation were investigated by comparing N2 growth factor-free medium to N2 media supplemented with different combinations of myogenic signaling regulators previously shown to enhance myotube differentiation (*Hicks et al., 2018*; *Tanoury, 2020*; *Selvaraj et al., 2019*). These media included N2 + SB medium, supplemented with a TGFβ inhibitor SB431542 (SB); N2 + SB + P + C medium, supplemented with SB, corticosteroid Prednisolone (P), and a GSK3 inhibitor/Wnt signaling activator, CHIR99021 (C); and N2+ SB + De + Da + F medium, supplemented with SB, the corticosteroid Dexamethasone (De), α gamma-Secretase/Notch signaling inhibitor DAPT (Da), and an adenyl cyclase activator Forskolin (F) (*Figure 4B*). All three supplemented N2 media significantly increased expression of *MYH7*, *MYH8*, and *CKM* in Ctrl and FSHD1 iMyotubes, but not adult *MYH1*, which as previously shown was expressed at high levels by bMyotubes except in N2+ SB + De + Da + F medium, which inhibited bMyoblast fusion and differentiation markers, but not *MYOD1* and *PAX3* expression (*Figure 4B* and *Figure 4— figure supplement 2B*). Increased muscle RNA expression in iMyotubes was correlated with increased networks of multinucleated myotubes, most prevalent in N2 + SB + P + C medium (*Figure 4B* and *Figure 4—figure supplement 2A*). The expression of myogenic regulators *MYOD1* and *PAX3* was variable, but their expression was not differentially affected by these media, showing that their effects are on differentiation and not myogenic commitment. By contrast, bMyoblast expression of muscle RNAs was increased in response to N2+ SB medium, particularly for FSHD1 bMyoblasts, likely by reducing DUX4-mediated toxicity. However, N2 + SB + P + C medium showed lower-level muscle RNA expression and N2 + SB + De + Da + F medium completely blocked muscle RNA expression and bMyoblast fusion (*Figure 4B* and *Figure 4—figure supplement 2B*). These findings reveal that iMyoblasts and bMyoblasts, and FSHD and Ctrl cells, respond differently to specialized differentiation media, reflecting their underlying differences in operative signaling mechanisms and toxicity responses.

## iMyoblast modeling of FSHD1 and its disease gene, DUX4

To investigate whether iMyoblasts have utility for human disease modeling, we compared expression of the FSHD disease gene, *DUX4,* in FSHD1 iMyoblasts and bMyoblasts. *DUX4* is a primate-specific member of the double homeobox (DUX) family of transcription factor genes of eutherian mammals, located in the D4Z4 retrotransposon repeat array near the telomere of chromosome 4 (*Gabriëls et al.,*

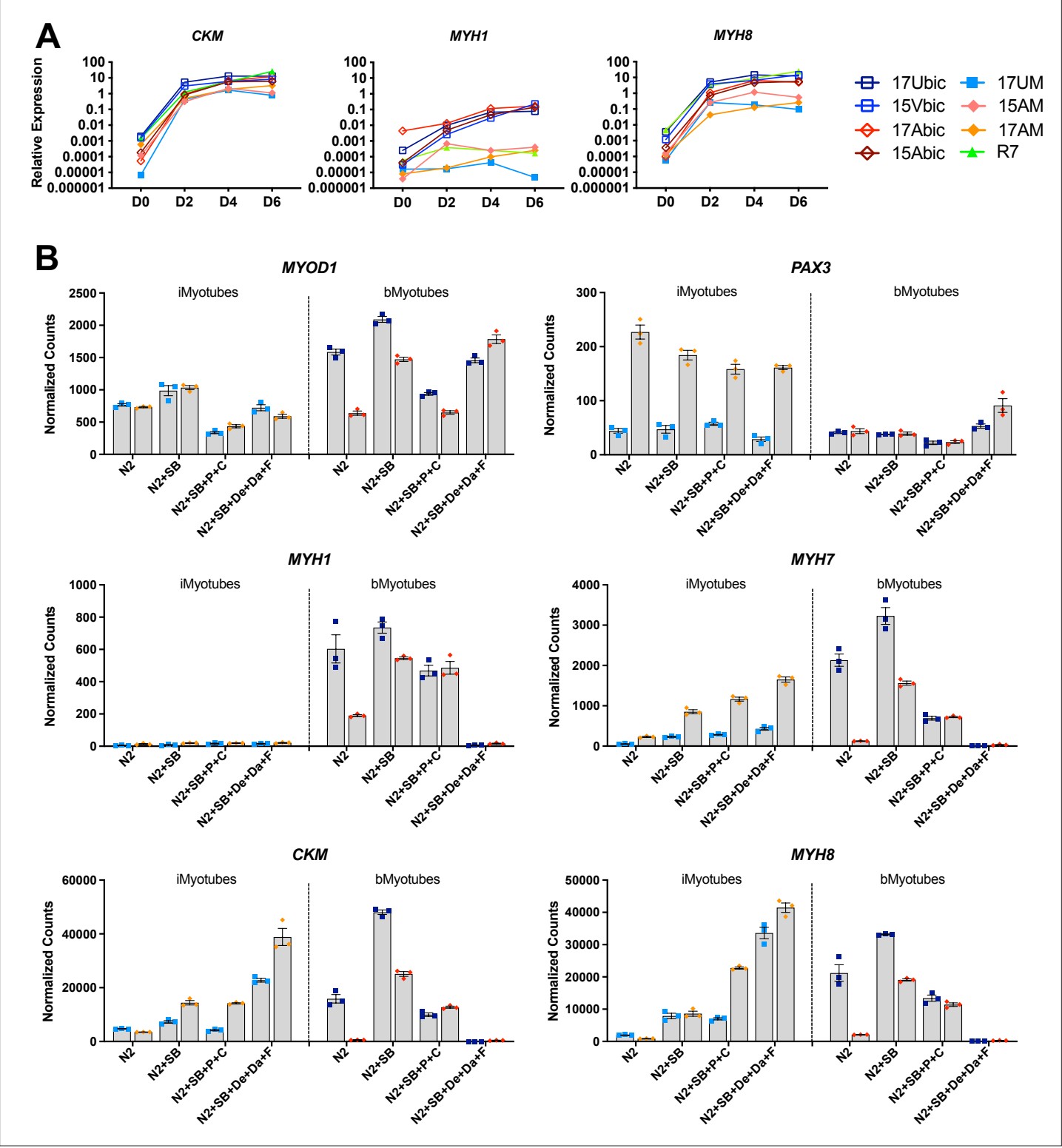

**Figure 4.** iMyoblasts upregulated muscle genes during ex vivo differentiation in response to specialized differentiation media. (**A**) Normalized qPCR assays of muscle RNAs *CKM*, *MYH1*, and *MYH8* in cultures of bMyoblasts and iMyoblasts from FSHD1, Ctrl, and LGMDR7 iPSCs during their differentiation for 6 days (D) in N2 serum-free medium. (**B**) NanoString digital RNA assays comparing the expression of *MYOD1*, *PAX3*, *MYH1*, *MYH7*, *MYH8*, and *CKM* in FSHD1 and Ctrl iMyotubes and bMyotubes of cohort 17 in response to N2 serum-free culture medium and N2 medium supplemented with signaling regulators as described in the text. NanoString digital counts were normalized to *RPL13A* and are shown on a linear scale.

*Figure 4 continued on next page*

*Figure 4 continued*

Each dot corresponds to an individual culture. Data are presented as mean ± SEM for each condition.

The online version of this article includes the following source data and figure supplement(s) for figure 4:

**Source data 1.** Source data for *Figure 4*.

**Figure supplement 1.** Effects of serum-free media and iPSC reprogramming on muscle and DUX4 target gene expression.

**Figure supplement 1—source data 1.** Source data for *Figure 4—figure supplement 1*.

**Figure supplement 2.** iMyotubes and bMyotubes respond differently to specialized differentiation media.

---

*1999*). *DUX4* developmentally functions to coordinate zygotic genome activation and male germline differentiation (*DeSimone et al., 2017*). In other tissues of healthy individuals, the *DUX4* locus is maintained in a highly condensed and CpG hypermethylated chromatin state and transcription of *DUX4* is repressed. FSHD is caused by *DUX4* misexpression in skeletal muscle in response to germline deletions and rearrangements that contract the D4Z4 locus on chromosome 4 to have ten or fewer repeats (FSHD1), or by mutations in chromatin-modifying genes such as *SMCHD1* and *DNMT3B* in combination with semi-short D4Z4 repeat lengths (FSHD2). These genetic disruptions lead to chromatin decondensation and CpG hypomethylation of the D4Z4 repeat locus, resulting in low-frequency DUX4 transcription that activates a battery of more than 100 DUX4-regulated germline target genes in FSHD1 bMyotubes nuclei and in FSHD1 patient muscle biopsies (*Geng et al., 2012*; *Snider et al., 2010*; *Lemmers et al., 2012*; *Yao et al., 2014*; *van den Boogaard et al., 2016*; *Lemmers et al., 2010*). Clinical disease requires that *DUX4* transcripts from the terminal D4Z4 unit be polyadenylated using a poly(A) site distal to the repeat array, associated with 'disease permissive' 4qA haplotypes (*Lemmers et al., 2010*). Misexpression of DUX4 and its target genes in muscles of FSHD patients leads to muscle toxicity and degeneration, resulting in clinical disease (*DeSimone et al., 2017*; *Lemmers et al., 2010*).

Expression of DUX4 and its target genes has previously been shown to be upregulated during the differentiation of patient biopsy-derived FSHD1 bMyoblasts, leading to myotube death (*Jones et al., 2012*; *DeSimone et al., 2017*; *Lemmers et al., 2010*). The expression of DUX4 target genes *MBD3L2*, *TRIM43*, *LEUTX*, and *ZSCAN4* was compared in Ctrl and FSHD1 iMyoblasts and bMyoblasts undergoing myotube differentiation in N2 serum-free differentiation medium. FSHD1 iMyoblasts and bMyoblasts both upregulated the expression of DUX4 target genes by >1000-fold compared to Ctrl iMyoblasts and bMyoblasts over 6 days of differentiation (*Figure 5A*). bMyoblast target gene upregulation was delayed by 24 hr compared to FSHD1 iMyoblasts following differentiation induction with N2 medium. However, iMyoblasts and bMyoblasts upregulated differentiation genes *MYH8*, *MYH1*, and *CKM* with similar kinetics, suggesting that *DUX4* transcription is more stringently repressed in bMyoblasts. DUX4 target gene expression was upregulated to similar levels in differentiating FSHD1 iMyoblasts derived from iPSCs reprogrammed from FSHD1 biopsy fibroblasts (*Figure 4—figure supplement 1B*) or FSHD1 biopsy bMyoblasts, indicating that parental somatic cell type used for iPSC reprogramming does not impact *DUX4* regulation during iMyoblast differentiation. DUX4 expression also was assayed in FSHD1 and Ctrl iMyoblasts undergoing myotube differentiation using a DUX4-GFP reporter (*Rickard et al., 2015*). FSHD1 iMyotubes expressed GFP in 4/100 nuclei, in contrast to undetectable expression in Ctrl iMyotubes (*Figure 5B*). These findings show that FSHD iMyotubes sporadically upregulate DUX4 in myotube nuclei as shown previously for bMyotubes using DUX4 IHC assays (*Chen et al., 2016*). However, nuclear frequency in iMyoblasts, as detected by the DUX4-GFP reporter, is 10× higher than in bMyoblasts, further indicating that *DUX4* transcription may be more stringently repressed in bMyoblasts.

The effects of media supplements on the expression of DUX4 target genes was assayed in FSHD1 and Ctrl cultures of iMyotubes, bMyotubes, and S3 iMyocytes using NanoString digital RNA assays (*Figure 5C*). Findings revealed that iMyotubes expressed highest levels of DUX4 target genes in growth factor-free N2 medium and less so in N2 + SB medium, whereas additional media supplements dramatically reduced the expression of DUX4 target genes, in contrast to their strong enhancement of muscle gene expression, uncoupling DUX4 target gene expression from expression of muscle differentiation genes (*Figures 4B and 5C*). bMyotubes expressed highest levels of DUX4 target gene and muscle RNAs in N2 + SB medium, less so in N2 and N2 + SB + P + C media, and not at all in N2 + SB + De + Da + F media which also blocked muscle RNA expression and bMyotube fusion (*Figure 4—figure supplement 2*). SB, a TGF-β inhibitor, optimally supports for DUX4 target gene

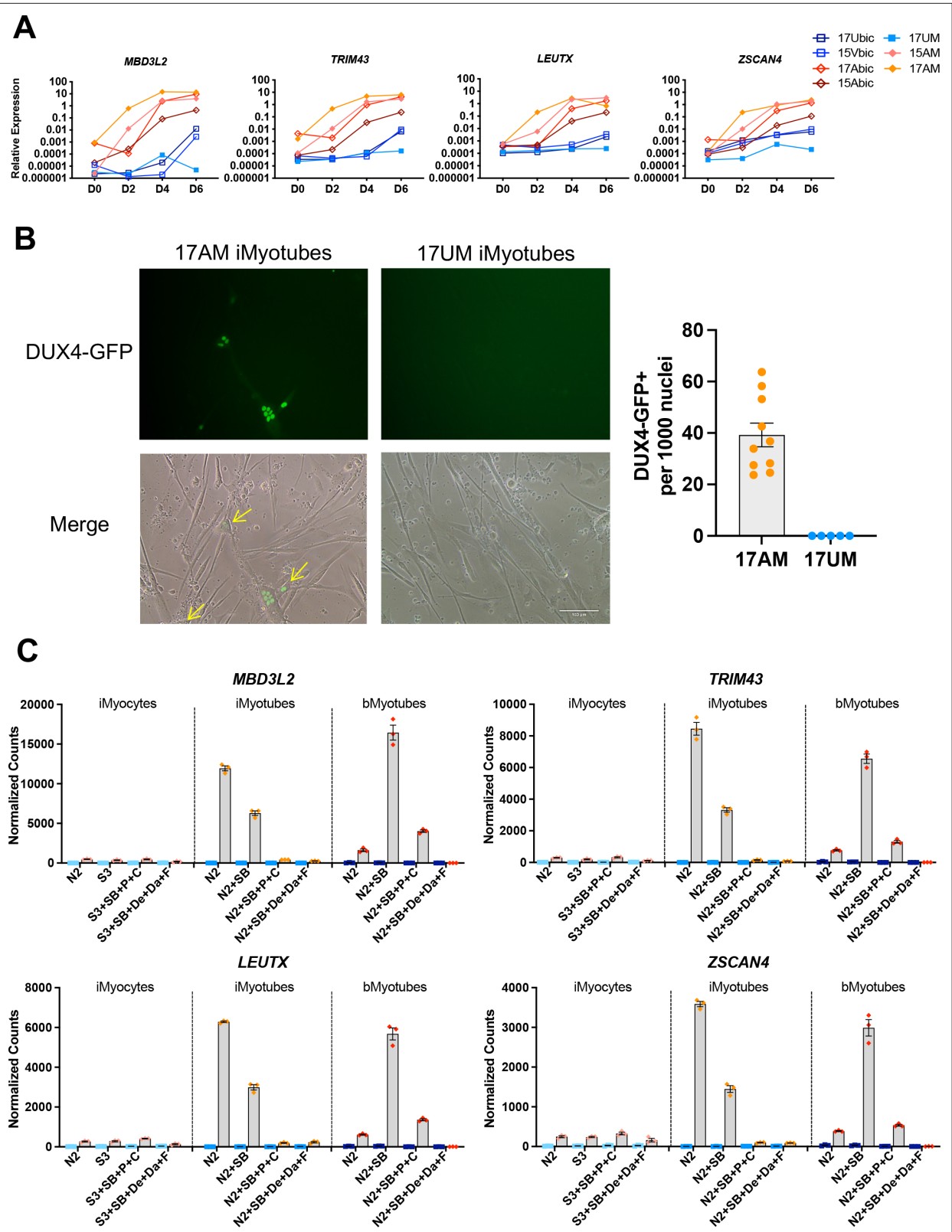

**Figure 5.** DUX4 and DUX4 target gene expression by iMyotubes, bMyotubes, and S3 iMyocytes in response to differentiation media. (**A**) Normalized qPCR assays of DUX4 target genes *MBD3L2*, *TRIM43*, *LEUTX,* and *ZSCAN4* in cultures of bMyoblasts and iMyoblasts of family cohorts 17 and 15 during differentiation for 6 days (D) in N2 serum-free medium. (**B**) DUX4-GFP reporter expression in FSHD1 bMyotubes (17Abic), FSHD1 iMyotubes (17AM), and Ctrl iMyotubes (17UM) after 7 days of differentiation. Scale bar=100 µm and quantification of DUX4 GFP+ nuclei/1000 nuclei/field is shown to the right.

*Figure 5 continued on next page*

*Figure 5 continued*

(**C**) NanoString digital RNA assays comparing the expression of DUX4 target gene RNAs in FSHD1 and Ctrl S3 iMyocytes, iMyotubes, and bMyotubes of cohort 17 in response to N2 serum-free culture medium and N2 medium supplemented with signaling regulators, as described in the text. NanoString digital counts were normalized to *RPL13A* and are shown on a linear scale. Each dot corresponds to an individual culture. Data are presented as mean ± SEM for each condition.

The online version of this article includes the following source data and figure supplement(s) for figure 5:

**Source data 1.** Source data for *Figure 5*.

**Figure supplement 1.** Losmapimod treatment decreases DUX4 target gene expression in FSHD1 iMyotubes.

**Figure supplement 1—source data 1.** Source data for *Figure 5—figure supplement 1*.

and muscle gene expression by FSHD bMyotubes, and N2 + SB + P + C and N2 + SB + De + Da + F media repressed DUX4 target gene expression in both iMyotubes and bMyotubes, likely through the inclusion of corticosteroids previously shown to repress DUX4 (*Pandey et al., 2015*). However, FSHD iMyoblasts and bMyoblasts were responsive to inhibition of DUX4 target gene expression by the p38 inhibitor, Losmapimod, currently in FSHD clinical trials (*Rojas, 2019*; *Figure 5—figure supplement 1*), showing that FSHD1 iMyotubes and bMyotubes share multiple pathways for *DUX4* regulation that are suitable for drug development targeting *DUX4* expression.

## Epigenetic repression of *DUX4* during FSHD iPSC reprogramming and iMyoblast selection

*DUX4* is an epigenetically regulated disease gene, which lead us to investigate whether iPSC reprogramming impacted *DUX4* epigenetic regulation. As shown above, FSHD1 iMyoblasts upregulated DUX4 target gene expression during differentiation similarly to adult biopsy FSHD1 bMyoblasts (*Figure 5A*, *Figure 6—figure supplement 1A*). However, we found that *DUX4* and its transcriptional target genes were not upregulated during S3 myocyte differentiation in response to specialized differentiation media (*Figure 5C*), or during earlier S1 and S2 stages of myogenic induction of both FSHD iPSCs and FSHD1 ESCs (*Figure 6A*, *Figure 6—figure supplement 1C*). *DUX4* and its target gene levels were higher in FSHD1 than Ctrl iPSCs and ESCs (*Figure 6A*, *Figure 6—figure supplement 1B*) but were 100-fold lower than FSHD1 iMyoblasts or bMyoblasts undergoing myotube differentiation. These findings contradict the earlier findings of *Caron et al., 2016*, who reported that *DUX4* is upregulated tenfold during S3 stage differentiation in one of the FSHD1 ESC lines also investigated in our study. Our findings do not exclude a low level of *DUX4* upregulation but it is small compared to the 1000-fold *DUX4* upregulation we observed during FSHD1 iMyotube differentiation (*Figure 6—figure supplement 1*).

To investigate whether the 4qA *DUX4* locus became methylated during iPSC reprogramming to repress *DUX4* expression, we performed bisulfite DNA sequencing (*Jones et al., 2014*). 4qA alleles of FSHD1 bMyoblasts from three FSHD family cohorts were hypomethylated (approximately 20% CpG methylation) before iPSC reprogramming whereas Ctrl iMyoblasts are hypermethylated (approximately 60% CpG methylation) (*Figure 6B*), as previously reported (*Jones et al., 2015a*). Since the uncontracted D4Z4 arrays for these two FSHD1 subjects have haplotypes not amplified in this assay (4qB and 4qA-L), methylation is specifically assayed only on contracted 4qA alleles. Our findings showed that DUX4 4qA remained hypomethylated in FSHD1 iPSCs and hypermethylated Ctrl iPSCs, and these methylation states were maintained throughout S1, S2, and S3 differentiation and in proliferating and differentiating iMyoblasts and iMyotubes (*Figure 6B and C*). DUX4 4qA alleles associated with D4Z4 contracted chromosomes of FSHD1 ESCs were also hypomethylated compared to Ctrl ESCs (*Figure 6—figure supplement 2*), as assayed using DUX4 bisulfite sequencing with 4qA-specific primers (*Jones et al., 2014*). These findings contradict a recent report, which found that DUX4 is hypermethylated in these same FSHD1 ESC lines (*Dion et al., 2019*). However, unlike the 4qA-specific primers we used, the bisulfite sequencing primers used in this earlier study can amplify all D4Z4 repeat units from both chromosomes, which we found obscures the hypomethylation at the distal 4qA repeat encoding DUX4 (data not shown). Unlike the DUX4 4qA locus, we found that the *MYOD1* core enhancer sequences of bMyoblasts became hypermethylated during iPSC reprogramming of parental bMyoblasts and then became demethylated during myogenic induction and differentiation (*Figure 6D*), concordant with *MYOD1* RNA upregulation (*Figure 1A*), as previously

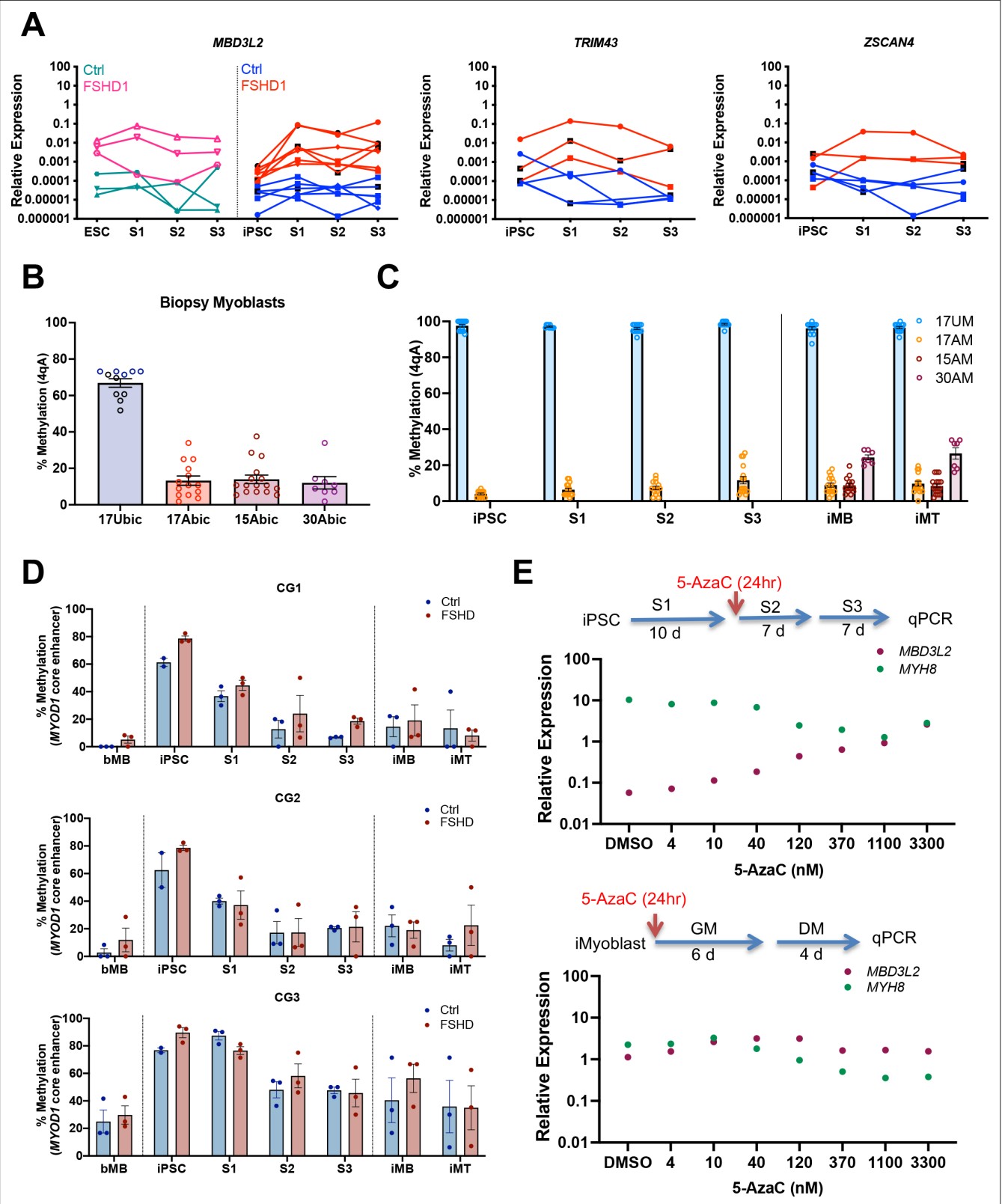

**Figure 6.** Epigenetic regulation of *DUX4* and *MYOD1* during iPSC reprogramming and iMyocyte and iMyotube differentiation. (**A**) Normalized qPCR assays of DUX4 target genes *MBD3L2*, *ZSCAN4,* and *TRIM43* during myogenic induction of FSHD1 and Ctrl ESC or iPSC. (**B**) Bisulfite sequencing of the *DUX4* 4qA locus in bMyoblasts from Ctrl (17Ubic) and FSHD1 subjects (15Abic, 17Abic, and 30Abic), shown as % 4qA CpG sites methylated. Each dot corresponds to the % CpG methylation of an individually sequenced DNA clone. (**C**) Bisulfite sequencing of *DUX4* 4qA alleles of iPSCs reprogrammed

*Figure 6 continued on next page*

*Figure 6 continued*

from parental bMyoblasts of Ctrl (17UM) and FSHD1 (17AM, 15AM, and 30AM) subjects, shown as % CpG methylation of *DUX4* 4qA alleles in iPSC, S1, S2, S3, and in iMyoblasts (iMB) and iMyotube (iMT) derived from these iPSC lines. (**D**) Bisulfite sequencing of *MYOD1* Core Enhancer, showing the % methylation of the three *MYOD1* core enhancer CpG sites (CG1, CG2, and CG3) from parental bMyoblasts (bMB) (15A, 17A, 30A, 15V, 17U, and 30W), reprogrammed iPSCs, cells at S1, S2, and S3 stages of primary myogenic induction, and iMyoblast (iMB) cell lines derived from these iPSC lines. Each dot corresponds to the average methylation of 10 sequenced DNA clones for each cell stage. (**E**) Normalized qPCR assays of the DUX4 target gene *MBD3L2* and the *MYH8* muscle RNA in S3 cultures (top panel) and iMyotube cultures (bottom panel) following treatment with increasing doses of 5-AzaC. Proliferating S2 cultures treated for 24 hr with 5-AzaC were cultured for 6 days in S2 growth medium followed by culture for 7 days in S3 differentiation medium for RNA isolation and qPCR. Proliferating cultures of iMyoblasts were treated for 24 hr with 5-AzaC and then cultured for 6 days in bMyoblast growth medium followed by culture for 4 days in Opti-MEM differentiation medium for RNA isolation and qPCR.

The online version of this article includes the following source data and figure supplement(s) for figure 6:

**Source data 1.** Source data for *Figure 6*.

**Figure supplement 1.** DUX4 expression by FSHD1 and Ctrl iPSCs and iMyoblasts and bMyoblasts.

**Figure supplement 1—source data 1.** Source data for *Figure 6—figure supplement 1*.

**Figure supplement 2.** CpG methylation of the 4qA allele of FSHD1 and Ctrl ESCs.

**Figure supplement 2—source data 1.** Source data for *Figure 6—figure supplement 2*.

observed in developing mouse embryos (*Brunk et al., 1996*). Therefore, methylation and demethylation machinery is operative in iPSCs and induced myogenic cells, but contracted 4qA alleles associated with FSHD1 are refractory to this machinery and changes in the methylation status of the 4qA DUX4 locus cannot account for the repression of DUX4 expression in FSHD1 iPSCs and ESCs and S3 iMyocytes.

To investigate whether *DUX4* repression during iPSC reprogramming is mediated by alternative epigenetic mechanisms, we screened a battery of epigenetic drugs for activation of DUX4 in S3 muscle cultures. The DNA demethylating drug 5-azacytidine (5-AzaC) effectively increased *DUX4* expression in FSHD1 S3 Myocyte cultures, in a concentration-dependent manner to levels comparable to those of iMyoblasts and bMyoblasts, as assayed by expression of the DUX4 target gene *MBD3L2* (*Figure 6E*). This finding shows that *DUX4* repression is mediated by the 5-AzaC sensitive DNA methylation of a *DUX4* regulatory locus that becomes inoperative in FSHD1 iMyoblasts and bMyoblasts.

## iMyoblast xenoengraftment and differentiation in mouse TA muscle

To investigate whether Ctrl and disease iMyoblasts xenoengraft and differentiate in vivo, FSHD1 and Ctrl iMyoblasts and bMyoblasts were engrafted into irradiated and BaCl$_2$ injured TA muscles of NSG immune-deficient mice (*Figure 7A*). Xenoengraftment was assayed by immunostaining with human-specific antibodies and by qPCR with human-specific qPCR primers. iMyoblast and bMyoblast xenografts were localized in humanized ECM domains within the mouse TA, as delineated by immunostaining with human-specific antibodies to lamin A/C, spectrin β1, and muscle collagen VI (*Figure 7B* and *Figure 7—figure supplement 1*). These domains were predominantly occupied by human nuclei and muscle fibers and were largely devoid of mouse nuclei and fibers, as identified by immunostaining with nuclear lamin A/C and sarcolemmal spectrin-β1 human-specific antibodies (*Figure 7B*). Variably sized spectrin β1+ fibers were detectable within 2 weeks following xenoengraftment and were associated with lamin A/C+ nuclei, with fibers oriented in parallel with residual peripheral mouse fibers along the sarcolemma matrix remaining after barium chloride destruction of mouse fibers. The numbers of spectrin β1+ fibers increased from 2 to 4 weeks and were higher in sections of bMyoblast xenografts than iMyoblast xenografts (*Figure 7B*, right). FSHD1 iMyoblast and bMyoblast xenografts had fewer spectrin β1+ fibers than Ctrl xenografts, likely reflecting *DUX4* and DUX4 target gene expression that results in fiber death (*Figure 7B*). FSHD1 and Ctrl iMyoblast and bMyoblast xenografts expressed muscle genes *MYH8* and *CKM* at comparable levels at 2 and 4 weeks post engraftment (*Figure 7C*), further validating muscle differentiation in xenografts. Muscle RNA expression was lower in FSHD1 xenograft muscles and xenografts were smaller and more variable in size (*Figure 7B and C* and *Figure 7—figure supplement 1*). Dystrophin+ mouse muscle fibers persisted in more peripheral regions of TA muscle (data not shown) and did not have centralized nuclei, validating the effectiveness of hindlimb irradiation for blocking mouse satellite cell regeneration. DAPI+/lamin A/C− mouse nuclei were also present within human muscle xenografts, but were not directly associated

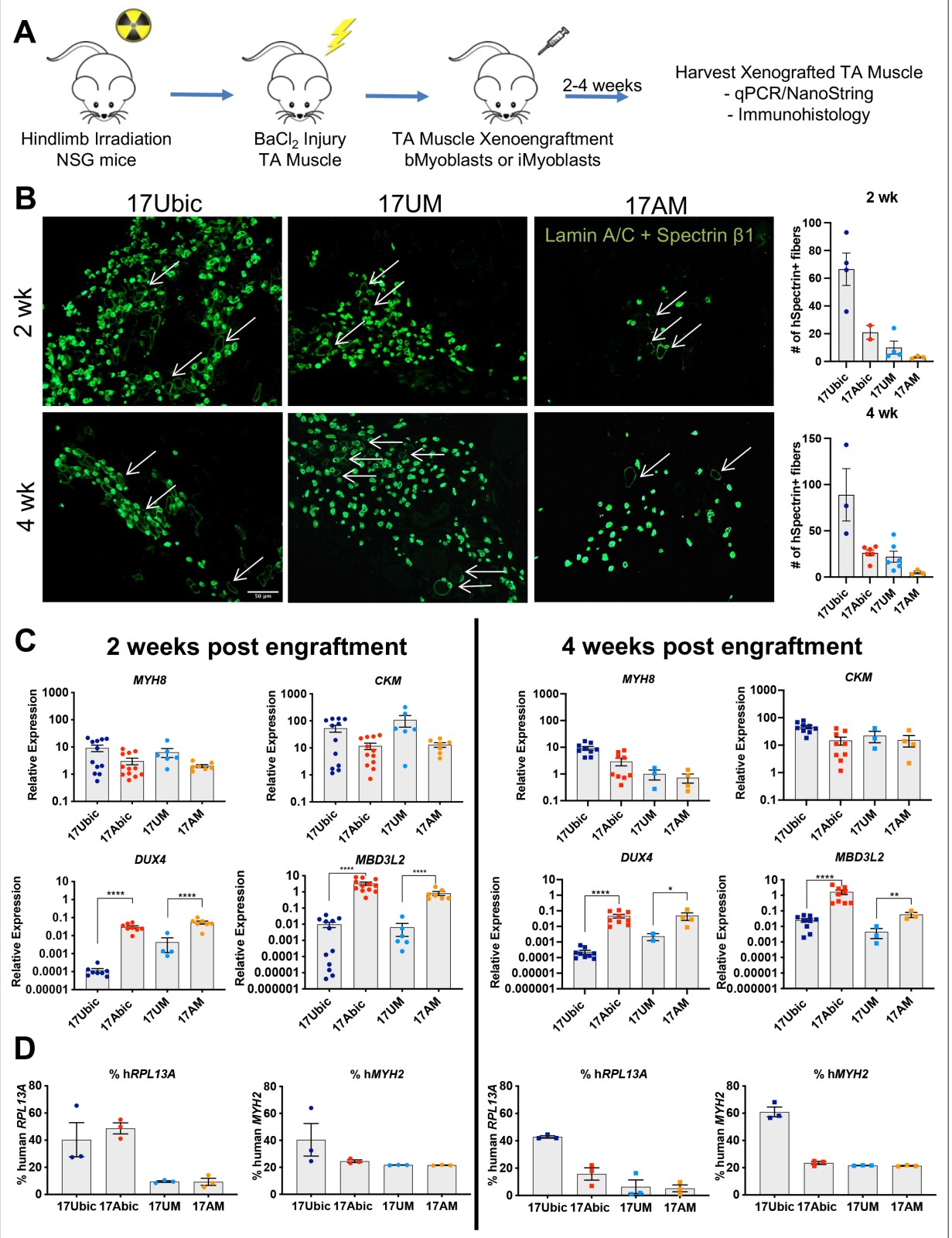

**Figure 7.** In vivo differentiation of FSHD1 and Ctrl iMyoblasts in muscle xenografts. (**A**) Schematic of muscle xenograft protocol. (**B**) Representative cryosections of 2 and 4 weeks Ctrl bMyoblasts (17Ubic) and Ctrl and FSHD1 iMyoblasts (17UM and 17AM) xenoengrafted TA muscles were immunostained with human-specific lamin A/C and spectrin-β1. Quantification for # of spectrin+ fibers for each condition is shown on the right. White arrows highlight human spectrin+ fibers. Scale bar=100 µm. (**C**) Normalized qPCR assays of the expression of *DUX4*, *MBD3L2*, *MYH8,* and *CKM*.

*Figure 7 continued on next page*

Figure 7 continued

*=p<0.05, **=p<0.01, ****=p<0.0001. (**D**) Percentage of human *RPL13A* and *MYH2* NanoString counts was calculated for each condition. Each dot corresponds to RNA from one xenografted mouse TA. Data are presented as mean ± SEM for each xenoengrafted condition.

The online version of this article includes the following source data and figure supplement(s) for figure 7:

**Source data 1.** Source data for *Figure 7*.

**Figure supplement 1.** bMyoblasts and iMyoblasts efficiently xenoengraft into irradiated and injured TA muscle of NSG mice.

with the sarcolemma of human fibers and included CD31+ mouse endothelial cells rebuilding the vascular system of xenograft muscle (data not shown). The engraftment efficiencies of iMyoblasts and bMyoblasts were assessed using NanoString digital RNA analysis of engrafted TA muscles by taking the ratio of human- and mouse-specific probes. Assays of human and mouse *RPL13A*, a housekeeping gene, showed that its expression in bMyoblast xenografts is 20–60% human, compared to 5–20% human in iMyoblast xenografts. Assays of human and mouse *MYH2*, a muscle gene, show that its expression in bMyoblast xenografts is 40–60% human, compared to 20–25% human in iMyoblast xenografts. The lower percentage of xenoengraftment using *RPL13A* assays likely reflects an abundance of non-muscle mouse cells in TA muscles compared to engrafted human cells, which are predominantly muscle. Overall, however, these findings show that iMyoblasts xenoengraft efficiently for investigations of muscle maturation and regeneration as described below. iMyoblasts and bMyoblasts differentially express a diversity of embryonic and adult ECM, matrix modifying, and regulatory genes that likely impact fetal-like iMyoblast xenoengraftment into the adult TA muscle (*Supplementary file 2*), providing a basis for enhancement of iMyoblast engraftment efficiencies.

## Myosin fiber type switching during maturation of iMyoblast muscle xenografts

To investigate whether iMyoblast xenograft muscle undergoes embryonic/fetal to adult fiber type maturation in the host mouse TA fast-twitch muscle (*Kammoun et al., 2014*), we compared expression of embryonic, adult fast, and adult slow Myosin Heavy Chain (MYH) isoforms (*Schiaffino et al., 2015*) ex vivo in cell culture and in vivo in xenografts using a custom NanoString panel of human-specific muscle RNA probes (*Figure 8*). Ex vivo, cultured FSHD1 and Ctrl iMyotubes and bMyotubes expressed high levels of *MYH3* (embryonic/fetal) isoform and intermediate levels of *MYH7* (cardiac/slow twitch) and *MYH8* (embryonic fetal/adult type 2A slow twitch) isoforms. iMyotubes expressed low levels of *MYH1* (late fetal/adult fast twitch 2×) and *MYH2* (fetal/adult fast twitch 2×) compared to their high expression in bMyotubes. However, 2 and 4 weeks iMyoblast xenografts downregulated expression of *MYH3*, *MYH7*, and *MYH8* isoforms as much as 100-fold, whereas expression of adult fast *MYH1* and *MYH2* isoforms was upregulated >100-fold. These findings show that iMyoblast xenografts undergo switching from embryonic/fetal to adult fast-twitch MYH isoforms. Similarly, but to a lesser extent, bMyoblast xenografts underwent MYH isoform switching, as evidenced by upregulation of *MYH1* and *MYH2*, and downregulation of *MYH3*, *MYH7*, and *MYH8* (*Figure 8*), while the expression of the *MYH4* isoform remained low and unchanged. These findings, therefore, establish that FSHD1 and Ctrl iMyoblast xenografts have regulatory plasticity in response to environmental signals from the adult host mouse TA fast-twitch muscle to promote fast MYH isoform switching and adult fast muscle maturation (*Wang and Kernell, 2001*).

## iMyoblast disease modeling of LGMDR7 and FKRP dystroglycanopathies

To further validate the utility of iMyoblasts for muscle disease research, we investigated the potential ex vivo and in vivo modeling applications of iMyoblasts from iPSCs derived from patients with FKRP dystroglycanopathies and LGMDR7 muscular dystrophy (*Figure 9*).

LGMDR7 muscular dystrophy is an autosomal recessive muscular dystrophy resulting from coding mutations of the *TCAP* locus encoding Telethonin, a skeletal and cardiac muscle myofibrillar protein that interacts with Titin to maintain sarcomere integrity. The pathogenic *TCAP* mutation in the patient cell line we used is an 8 bp microduplication in the Telethonin coding sequence (*Cotta et al., 2014*). LGMDR7 iMyoblasts undergo muscle differentiation and expression of muscle genes ex vivo (*Figure 4A*) and in muscle xenografts (*Figure 9A and B*). Previously we showed that Telethonin

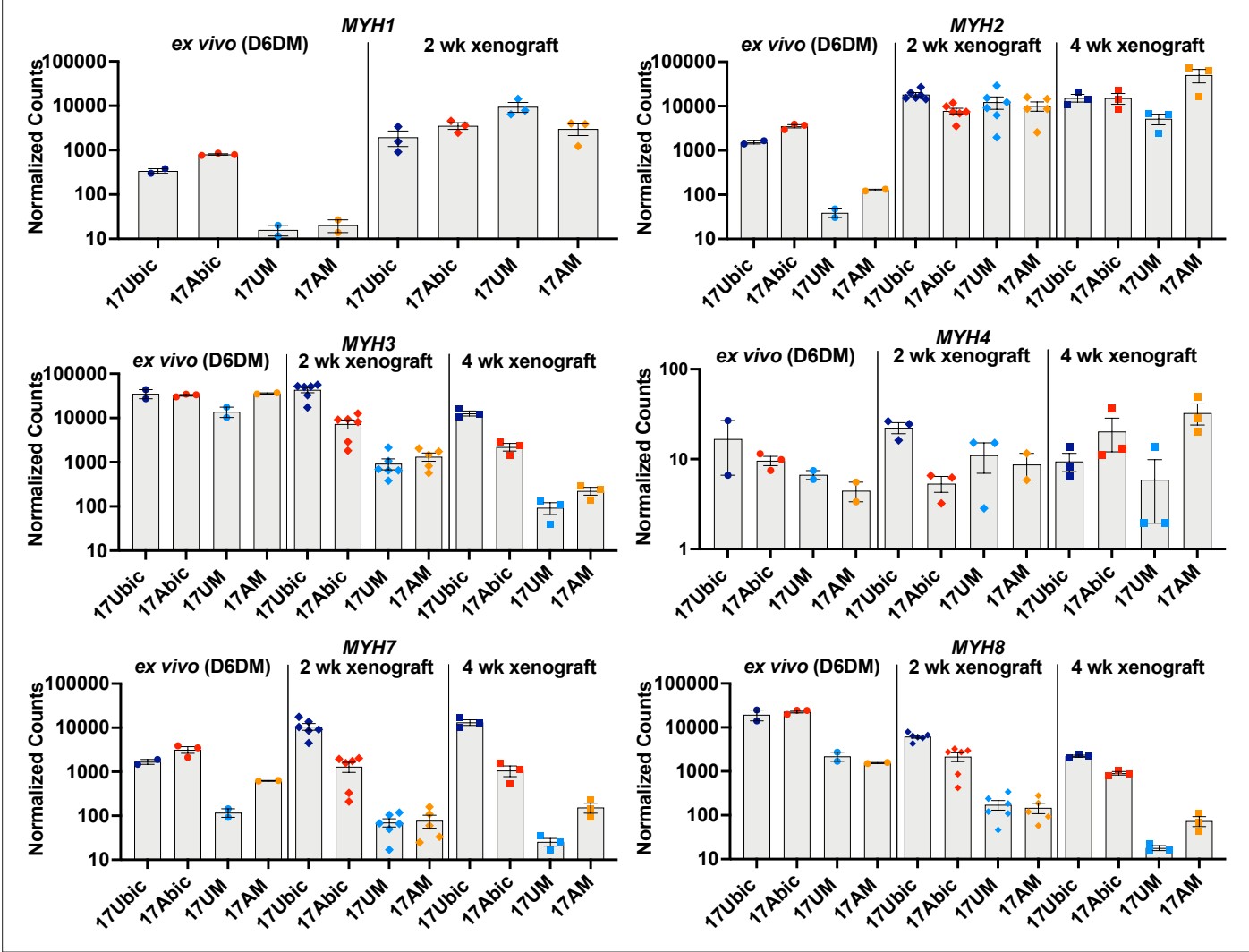

**Figure 8.** Embryonic/adult MYH isoform switching during maturation of iMyoblast and bMyoblast muscle xenografts. NanoString digital assays of the expression of MYH isoforms in cultures of FSHD1 (17Abic) and Ctrl (17Ubic) bMyotubes and FSHD1 (17AM) and Ctrl (17UM) iMyotubes at Day 6 differentiation in Opti-MEM (D6DM), compared to MYH isoform expression in 2 and 4 weeks TA muscle xenografts of these same cell-lines. Digital counts were normalized to a panel of four human-specific housekeeping genes and plotted on a $log_{10}$ scale. Each dot corresponds to RNA from one xenoengrafted mouse TA muscle (2 and 4 weeks xenograft) or one ex vivo bMyotube or iMyotube culture. Data are presented as mean ± SEM for each cell or xenograft sample.

The online version of this article includes the following source data for figure 8:

**Source data 1.** Source data for **Figure 8**.

expression could be efficiently restored in LGMDR7 iMyoblasts by microhomology-mediated repair using Cas9 and *TCAP* guide RNAs targeting the microduplication (*Iyer et al., 2019*), establishing the utility of iMyoblasts for CRISPR disease gene editing therapeutics. Additionally, assays of the efficiencies of xenoengraftment showed that S1 stage FSHD1 cells engrafted poorly and did not differentiate well compared to iMyoblast xenografts, as evidenced by qPCR assays for muscle RNAs showing high raw Ct values comparable to unengrafted mouse TA samples (*Figure 9C*). These data further highlight the efficiency of iMyoblast engraftment and their utility for in vivo studies of muscle maturation and regeneration.

WWS and LGMDR9 (formerly LGMD2I) FKRP dystroglycanopathies are caused by recessive mutations in the coding sequences of the gene *FKRP*, disrupting its function in the glycosylation of α-Dystroglycan (α-DG) for laminin binding to maintain cellular interactions with the ECM (*Piccolo et al., 2002*). WWS subjects carry loss-of-function *FKRP* mutations that fully disrupt α-DG glycosylation for

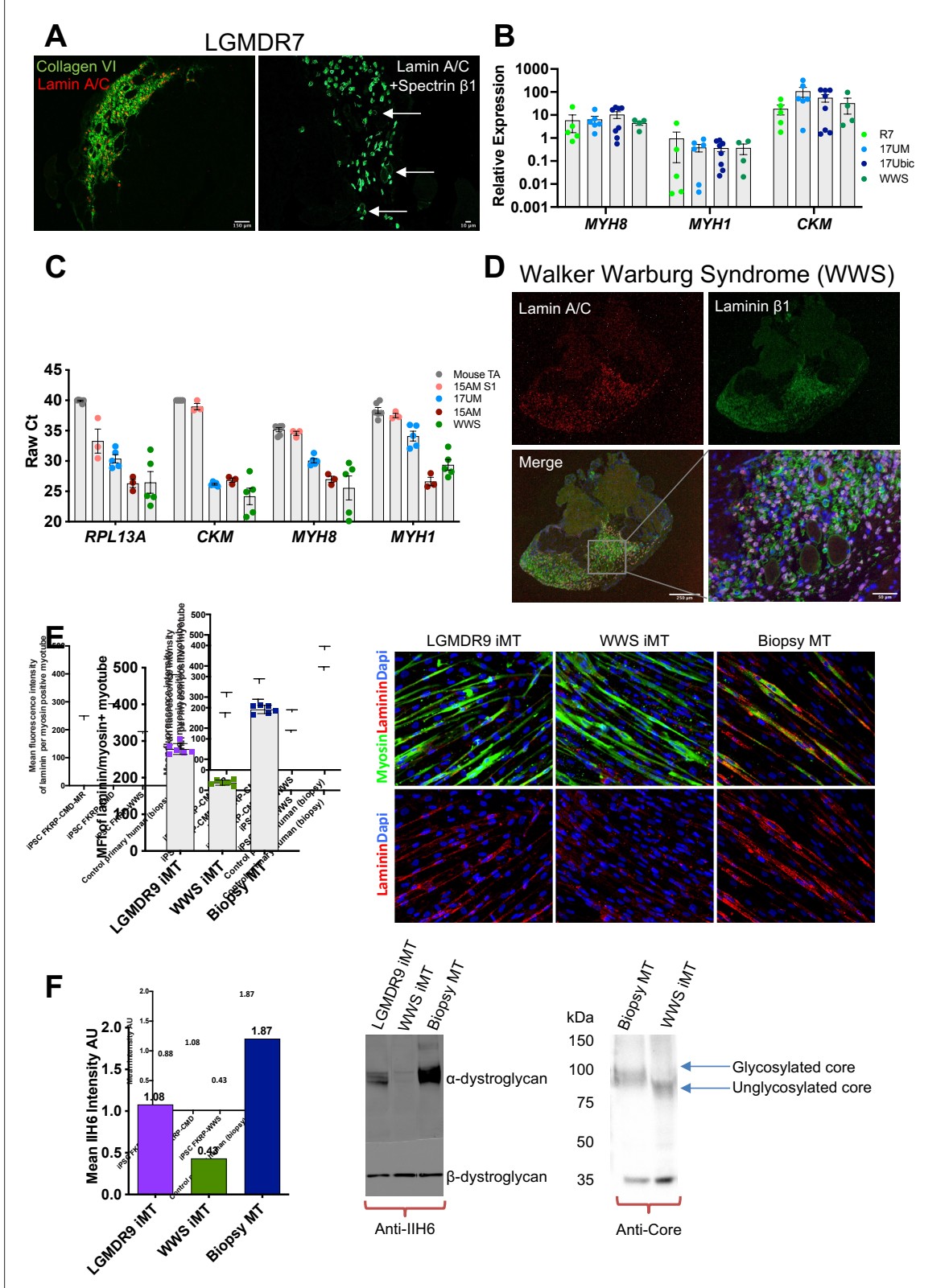

**Figure 9.** iMyoblast modeling LGMDR7 and FKRP dystroglycanopathies. (**A**) Representative images from LGMDR7 muscle xenograft cryosections immunostained with human-specific antibodies (left) lamin A/C (red) and collagen VI (green) or (right) lamin A/C and spectrin β1 (green). Scale bar=150 μm and 10 μm. Arrows identify spectrin+ fibers. (**B**) Normalized qPCR assays of muscle RNAs 3 weeks after engraftment of bMyoblasts (17Ubic) or iMyoblasts (17UM, LGMDR7, WWS). Each dot corresponds to RNA from one xenografted mouse TA. Data are presented as mean ± SEM. (**C**) Raw

*Figure 9 continued on next page*

*Figure 9 continued*

Ct values comparing engraftment of S1 cells to Ctrl, FSHD1, LGMDR7, and WWS iMyoblasts and control unengrafted mouse TA using human-specific *RPL13A*, *CKM*, *MYH8*, and *MYH1*. (**D**) Representative images of cryosections of WWS iMyoblast engrafted TA xenograft muscles immunostained with human lamin A/C and human laminin β1. Scale bar=250 μm, inset scale bar=50 μm. (**E**) Representative images from bMyotube (biopsy MT) or FKRP disease iMyotube cultures (LGMDR9 iMT and WWS iMT) immunostained with laminin (red) and myosin (green) antibodies, and DAPI (blue). The mean fluorescent intensity averaged from multiple fields of view for each condition is shown on the left. Data are presented as mean ± SD. (**F**) Western blot assays of the expression of α-DG and β-DG in biopsy MTs, WWS, and LGMDR9 iMyotubes (iMT) using glycosylation-specific IIH6 antibody and Core Dystroglycan antibody. Mean intensity of α-DG for IIH6 is shown on the left.

The online version of this article includes the following source data for figure 9:

**Source data 1.** Source data for *Figure 9*.

**Source data 2.** Source data for *Figure 9F*.

laminin binding, causing clinically severe neuronal and muscle developmental damage and death following birth. LGMDR9 FKRP mutations cause partial loss of FKRP enzymatic function, resulting in adolescent onset of muscle weakness without neuronal involvement, and partial disruption of α-DG glycosylation and laminin binding. WWS iMyoblasts xenoengraft efficiently into irradiated and injured TA muscles of NSG mice, as evidenced by IF assays using human-specific lamin A/C and laminin β1 antibodies, and express muscle RNAs at similar abundance to Ctrl iMyoblasts, FSHD1 iMyoblasts, and LGMDR7 iMyoblasts (*Figure 9B and D*). Ex vivo laminin binding assays of fluorescently labeled laminin to WWS iMyotubes showed almost complete absence of binding compared to Ctrl Myotubes, and LGMDR9 iMyotubes showed intermediate binding (*Figure 9E*), reflecting the dose-dependent FKRP enzymatic loss of function. Similarly, α-DG glycosylation was assayed biochemically by probing Western blots of iMyotube membrane extracts with IIH6 antibody, which reacts with α-DG glycosylation epitopes, and with α-DG core protein antibody that assays glycosylation based on the reduced mobility of α-DG on SDS gels. WWS iMyotubes were nearly completely deficient in IIH6 reactivity and had smaller, unglycosylated α-DG core protein (75 kDa) compared to glycosylated Ctrl iMyotubes (100 kDa), whereas LGMDR9 had intermediate IIH6 intensity (*Figure 9F*), showing that differences in glycosylation between WWS and the less severe LGMDR9 alleles can model the functional and clinical severity of FKRP mutations. Taken together, these data demonstrate the capacity of iMyoblasts in ex vivo and in vivo assays and provide well-developed models for FKRP therapeutic development as well as for muscle regeneration studies, as described below.

## Regenerative potential of iMyoblast muscle xenografts

To investigate whether iMyoblast xenograft muscle had regenerative potential, primary xenografts were generated by xenoengraftment of Ctrl and WWS iMyoblasts and assayed by immunostaining for PAX3 expressing cells (*Figure 10A*). Both Ctrl and WWS primary xenografts had a significant population of PAX3+ nuclei in humanized regions of the engrafted TA muscle, indicating that xenografts maintain a progenitor cell population. PAX3+ nuclei co-stained with human-specific lamin A/C and were associated with human muscle fibers as detected by immunostaining with human-specific spectrin-β1 whereas other nuclei appeared to be localized more interstitially. Additionally, xenografts included lamin A/C+ nuclei that did not immunostain with PAX3 (*Figure 10B*). To investigate whether xenografts have regenerative potential, we induced a secondary barium chloride injury in WWS iMyoblast engrafted muscle. PAX3+/lamin A/C+ nuclei were observed in humanized muscle after secondary injury (*Figure 10B*).

WWS iMyoblast secondary injury xenografts were assayed for differentiated muscle fibers using human-specific neonatal myosin heavy chain, collagen VI, and laminin β1 staining identifying human muscle fibers in humanized areas of injury and regeneration (*Figure 10C*). These findings support the conclusion that iMyoblast xenografts maintain a population of PAX3+ myogenic cells that function to regenerate human muscle in response to injury.

## Discussion

Here, we report the invention of an efficient and reliable method to isolate human *PAX3*-expressing muscle stem cells, iMyoblasts, from iPSCs of Ctrl and muscular dystrophy patients using transgene-free iPSC myogenic induction in combination with reserve stem cell selection, which has not been

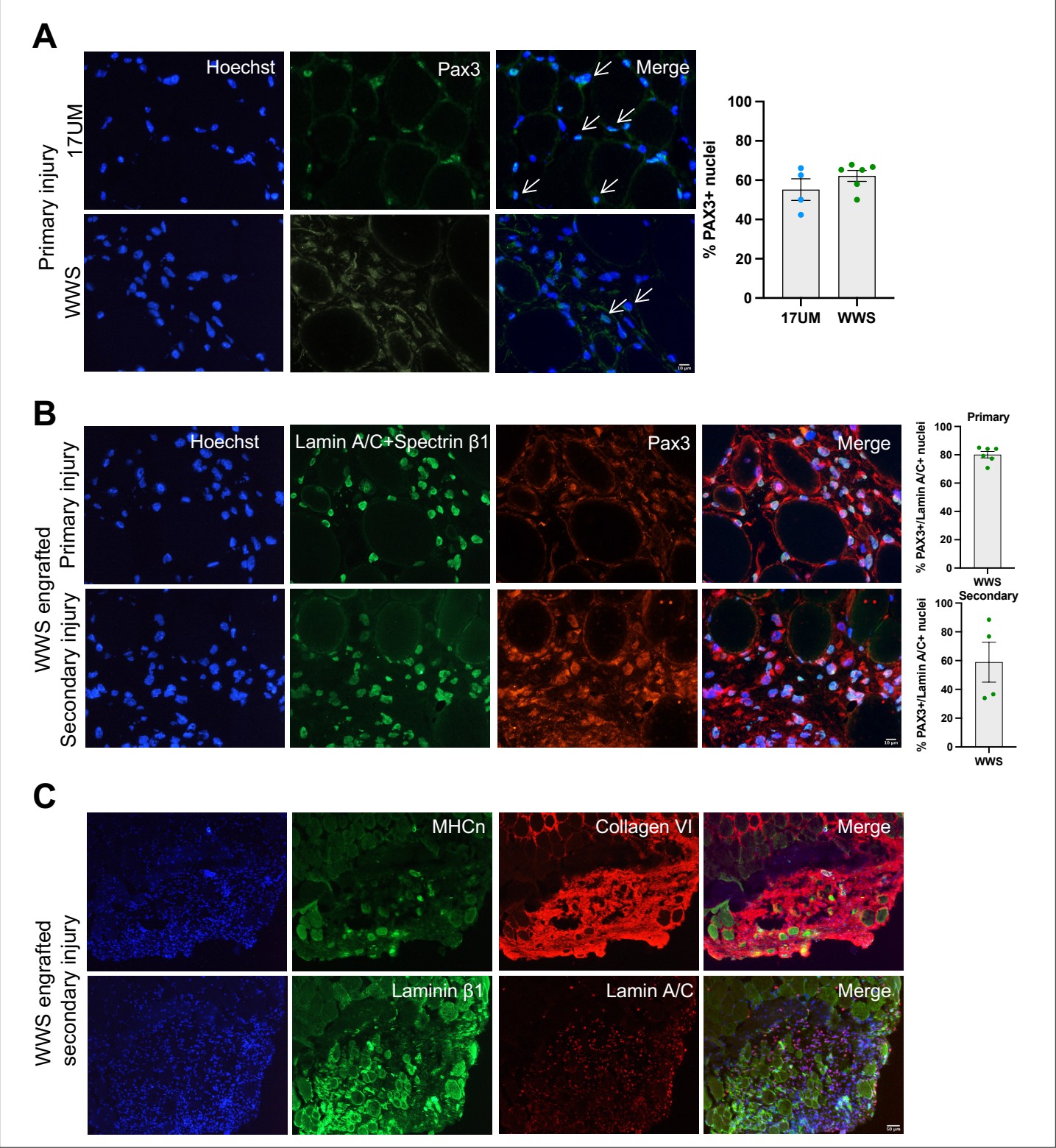

**Figure 10.** iMyoblasts have regenerative potential after secondary muscle injury. (**A**) Representative images from 17UM and WWS muscle xenograft cryosections immunostained with PAX3 and stained with Hoechst. The percentage of PAX3+ nuclei is shown on the right. N=4, 17UM engrafted sections and 6, WWS engrafted sections. Data are presented as mean ± SEM for each engraftment condition. Scale bar=10 μm. (**B**) Representative images from primary and secondary injury WWS muscle xenograft cryosections immunostained with PAX3 (red), lamin A/C and spectrin-β1 (green), and stained with Hoechst. N=6, primary injury WWS engrafted sections and 4, secondary injury WWS engrafted sections. Data are presented as mean ± SEM for each

*Figure 10 continued on next page*

*Figure 10 continued*

engraftment condition. Scale bar=10 μm. (**C**) Representative images from secondary injury WWS muscle xenograft cryosections immunostained with human-specific (TOP) neonatal myosin heavy chain (MHCn) (green), collagen VI (red) and stained with Hoechst or (BOTTOM) laminin β1 (green), lamin A/C (red) and stained with Hoechst. Scale bar=50 μm.

The online version of this article includes the following source data for figure 10:

**Source data 1.** Source data for *Figure 10*.

previously utilized for isolation of iPSC-derived muscle progenitors. Other iPSC myogenesis protocols have been optimized to generate differentiated skeletal muscle from mouse and human ESCs and iPSCs, either by transgene misexpression of muscle master regulatory genes including *MYOD1* (*Dekel et al., 1992*; *Maffioletti et al., 2015*) and *PAX7* (*Darabi et al., 2012*; *Rao et al., 2018*), or by transgene-free protocols that transition ESCs and iPSCs through an early developmental progression that regulates skeletal myogenesis in the vertebrate embryo (*Caron et al., 2016*; *Chal et al., 2015*; *van der Wal et al., 2018*; *Hicks et al., 2018*) rather than through direct induction of PAX3-expressing muscle progenitors as in our protocol. Our approach also differs in its efficient reserve cell selection of highly enriched, stably committed, and expandable populations of iMyoblasts by growth selection, without requiring FAC sorting of induced subpopulations. Our findings demonstrate the utility of our protocol for the isolation of iMyoblasts from iPSCs from healthy control (Ctrl) subjects and patients with FSHD1, LGMDR7, WWS, and LGMDR9 muscular dystrophies that express the molecular pathologies of these diseases. iMyoblasts can be maintained and expanded for at least 30 population doublings while stably retaining their capacity for myogenic differentiation. iMyoblast cell expansion enables statistically powered cell and molecular studies of human myogenesis and investigations of muscle disease pathology and therapeutic development using both ex vivo and in vivo in muscle xenograft models, as demonstrated in this study. By contrast, adult bMyoblasts often cannot be propagated from biopsies of muscular dystrophy patients who have advanced muscle pathology or have limited growth in culture because of the age and disease-related senescence (*Webster and Blau, 1990*). The advantages of iPSC derived iMyoblasts for mechanistic studies of epigenetic disease mechanisms are shown by our discovery of a previously unknown developmental epigenetic repression of the FSHD1 disease gene, *DUX4*. Finally, iMyoblasts undergo enhanced differentiation in response to specialized media, which enable 3D modeling of muscle maturation and contractile and electrophysiological activities that will further enhance knowledge of disease mechanisms and enable therapeutic development. iMyoblasts express markers of fetal myoblasts, including *PAX3* and *CD82*, as well as *NFIX*, a regulator of the transcriptional switch from embryonic to fetal myoblasts (*Messina et al., 2010*). The iMyoblast transcriptome is distinct from iPSC-induced S1 and S2 myogenic progenitors, which do not express *NFIX* and differentiate as mononucleated myocytes typical of the earliest stage embryonic muscle, and also is distinct from adult bicep muscle bMyoblasts, which express *PAX7*, *MYF5*, *MRF4* as well as *PAX3* and *MYOD1*, and higher levels of *NFIX*. The iMyoblast transcriptome also is distinct from the transcriptomes of iPSC-induced *PAX7* Myoblasts, but share the fetal-like gene expression profiles along the developmental trajectory of limb myogenic cells described by *Xi et al., 2020*. iMyoblasts also may be related to adult *PAX3* myogenic cells identified in muscles such as diaphragm or to stress resistant *PAX3* myogenic cells (*Der Vartanian et al., 2019*; *Scaramozza et al., 2019*), noting that, in humans, *PAX3* and *PAX7* may be functionally redundant, as suggested by recent genetic studies showing that *PAX7* is not an essential gene for satellite cell production and muscle regeneration (*Marg et al., 2019*). In addition to *PAX3* and *MYOD1*, iMyoblasts express an array of other transcription factors, including *GATA3*, *GATA6*, *HAND2*, *MEIS2*, *GLI2*, *NOTCH1*, *ETS1*, and *ETS2*, that most certainly play a role to regulate the isoform-specific expression of ECM, focal adhesion, migration/chemotaxis, and cell signaling genes of the iMyoblast transcriptome.

The fetal-like phenotype of iMyoblasts ex vivo is further reflected by the expression of embryonic/fetal *MYH8* during iMyotube differentiation, in contrast to *MYH1* expression by adult bMyoblasts. Significantly, iMyoblast xenoengraftment into the mouse TA muscle generated muscle that downregulated expression of embryonic *MYH8* and upregulated expression of adult fast *MYH1* during fiber differentiation, demonstrating the transcriptional plasticity of iMyoblasts to respond to in vivo signals in the fast fiber type TA muscle to mediate adult MYH isoform switching. The signals in xenograft muscle that initiate adult switching may be controlled by physiological mechanisms such as innervation (*Xi et al., 2020*) or by conversion of engrafted iMyoblasts to a more adult-like bMyoblast

progenitor phenotype. In either case, the capacity of iMyoblasts to engraft directly into adult muscle to produce xenograft muscle that undergoes adult MYH isoform switching in combination with the capacity of iMyoblasts for extensive cell expansion and efficient CRISPR gene editing (*Iyer et al., 2019*) supports their potential utility for development of muscle stem cell therapies, which to date have been unsuccessful. iMyoblasts have growth and differentiation characteristics of self-renewing stem cells based on their dual commitment to differentiation and iMyoblast reserve cell renewal ex vivo and the capacity of iMyoblast xenograft muscles to regenerate human muscle and PAX3+ cells in response to secondary injury. The stem cell characteristics of iMyoblasts likely reflect their isolation using reserve cell selection from cultures of highly differentiated S3 stage muscle following transgene-free induction using the myogenic induction protocol, which was optimized to generate differentiated cultures of S3 iMyocytes without transitioning iPSCs through earlier mesodermal and somite developmental stages (*Caron et al., 2016*). A realized consequence was that this differentiation protocol also generated reserve cells that could be isolated from differentiated cultures by growth factor activation. The mechanisms that control reserve cell generation are not yet understood, but likely are based on the initiation of *MYOD1* autoregulation in earlier stage cells during the S1 and S2 stages of iPSC induction, as shown in previous studies of *MYOD1* overexpression in somatic cells (*Weintraub et al., 1989*). iMyoblasts maintain low level *MYOD1* and *PAX3* expression cell-autonomously, which enables their myogenic potential over 30 population doublings while retaining their commitment to differentiate as iMyotubes and replenish *PAX3* iMyoblasts as reserve cells. In vivo, this enables iMyoblasts to xenoengraft and generate differentiated muscle and a population of *PAX3* expressing cells that are retained in xenografts in response to secondary injury. The regulatory mechanisms that control the distinct growth, differentiation and self-renewal capacities of iMyoblasts are not yet understood. iMyoblasts express genes well-known to regulate *MYOD1* function, myoblast differentiation and self-renewal, including *ID* genes (*Benezra et al., 1990*), Wnt and BMP antagonists *DKK1* ( *Jones et al., 2015b*) and *GREM1* (*Fabre et al., 2020*), and *RGS4* (*Yilmaz et al., 2016*), *TXNRD1* (*Mercatelli et al., 2017*), *UCHL1* (*Gao et al., 2017*), NR2F2 (*Lee et al., 2017*), *FOXC2* (*Lagha et al., 2009*), and *HMGA2* and *IGF2BP* myogenic regulators (*Li et al., 2012*). Studies to investigate the myogenic and stem cell functions of these iMyoblast genes and others identified in our studies are now approachable using CRISPR gene editing.

Overall, our findings establish a protocol for isolation of iMyoblasts and establish its suitability for ex vivo and in vivo investigations of human myogenesis and muscle disease pathogenesis. Our expectation is that iMyoblasts will enable development and validation of multiple disease corrective modalities, including gene editing and muscle stem cell therapeutics, to ameliorate disease pathology and disabilities associated with muscular dystrophies.

# Materials and methods

### Key resources table

| Reagent type (species) or resource | Designation | Source or reference | Identifiers | Additional information |
|---|---|---|---|---|
| Strain, strain background (*Mus musculus*) | NOD.Cg-*Prkdc*^scid^*IL2ry*^tmiWjl^/SzJ | Jackson Lab | Stock No: 005557 | |
| Cell line (*Homo sapiens*) | 15Abic biopsy myoblast | *Homma et al., 2012*; *Jones et al., 2012* | | |
| Cell line (*H. sapiens*) | 15Vbic biopsy myoblast | *Homma et al., 2012*; *Jones et al., 2012* | | |
| Cell line (*H. sapiens*) | 17Abic biopsy myoblast | *Jones et al., 2012* | | |
| Cell line (*H. sapiens*) | 17Ubic biopsy myoblast | *Jones et al., 2012* | | |
| Cell line (*H. sapiens*) | 30Abic biopsy myoblast | *Jones et al., 2012* | | |

*Continued on next page*

*Continued*

| Reagent type (species) or resource | Designation | Source or reference | Identifiers | Additional information |
|---|---|---|---|---|
| Cell line (*H. sapiens*) | 30Wbic biopsy myoblast | University of Massachusetts Medical School https://www.umassmed.edu/wellstone/ | | |
| Cell line (*H. sapiens*) | 15AM iPSCs | This paper | | 15AM iPSCs were reprogrammed from 15Abic biopsy CD56+ myoblast at UMMS |
| Cell line (*H. sapiens*) | 15VM iPSCs | This paper | | 15VM iPSCs were reprogrammed from 15Vbic biopsy CD56+ myoblast at UMMS |
| Cell line (*H. sapiens*) | 17AM iPSCs | This paper | | 17AM iPSCs were reprogrammed from 17Abic biopsy CD56+ myoblast at UMMS |
| Cell line (*H. sapiens*) | 17AF iPSCs | This paper | | 17UM iPSCs were reprogrammed from 17Abic biopsy CD56- fibroblast at UMMS |
| Cell line (*H. sapiens*) | 17UM iPSCs | This paper | | 17UM iPSCs were reprogrammed from 17Ubic biopsy CD56+ myoblast at UMMS |
| Cell line (*H. sapiens*) | 30AM iPSCs | This paper | | 30AM iPSCs were reprogrammed from 30Abic biopsy CD56+ myoblast at UMMS |
| Cell line (*H. sapiens*) | 30WM iPSCs | This paper | | 30WM iPSCs were reprogrammed from 30Wbic biopsy CD56+ myoblast at UMMS |
| Cell line (*H. sapiens*) | 54574/75 iPSCs | This paper | | 54574/75 iPSCs were reprogrammed from skin fibroblast at UMMS |
| Cell line (*H. sapiens*) | 54585 iPSCs | This paper | | 54585 iPSCs were reprogrammed from skin fibroblast at UMMS |
| Cell line (*H. sapiens*) | LGMDR7 iPSCs | Formerly LGMD2G (*Iyer et al., 2019*) | | |
| Cell line (*H. sapiens*) | LGMDR9 FP iPSCs | This paper | | FP iPSCs were reprogrammed from skin fibroblast at Sanford Burnham Prebys Medical Discovery Institute |
| Cell line (*H. sapiens*) | WWS iPSCs | This paper | | WWS iPSCs were reprogrammed from skin fibroblast at Sanford Burnham Prebys Medical Discovery Institute |
| Cell line (*H. sapiens*) | 15AM iMyoblasts | This paper | | 15AM iMyoblasts were made from 15AM iPSCs |
| Cell line (*H. sapiens*) | 15VM iMyoblasts | This paper | | 15VM iMyoblasts were made from 15VM iPSCs |
| Cell line (*H. sapiens*) | 17AM iMyoblasts | This paper | | 17AM iMyoblasts were made from 17AM iPSCs |
| Cell line (*H. sapiens*) | 17AF iMyoblasts | This paper | | 17AF iMyoblasts were made from 17AF iPSCs |
| Cell line (*H. sapiens*) | 17UM iMyoblasts | This paper | | 17UM iMyoblasts were made from 17UM iPSCs |
| Cell line (*H. sapiens*) | 30AM iMyoblasts | This paper | | 30AM iMyoblasts were made from 30AM iPSCs |
| Cell line (*H. sapiens*) | 30WM iMyoblasts | This paper | | 30WM iMyoblasts were made from 30WM iPSCs |
| Cell line (*H. sapiens*) | 54574/75 iMyoblasts | This paper | | 54574/75 iMyoblasts were made from 30WM iPSCs |
| Cell line (*H. sapiens*) | 54585 iMyoblasts | This paper | | 54585 iMyoblasts were made from 54585 iPSCs |
| Cell line (*H. sapiens*) | LGMDR7 iMyoblasts | This paper | | LGMDR7 iMyoblasts were made from LGMDR7 iPSCs |
| Cell line (*H. sapiens*) | LGMDR9 FP iMyoblast | This paper | | LGMDR9 FP iMyoblasts were made from LGMDR9 iPSCs |

*Continued on next page*

*Continued*

| Reagent type (species) or resource | Designation | Source or reference | Identifiers | Additional information |
|---|---|---|---|---|
| Cell line (*H. sapiens*) | WWS iMyoblasts | This paper | | WWS iMyoblasts were made from WWS iPSCs |
| Antibody | MyoD1 (Clone: 5.8A) (Mouse Monoclonal) | Dako | Cat #: M3512 | IF (1:50) |
| Antibody | MF20 (Mouse Monoclonal) | DSHB | Cat #: AB_2147781 | IF (1:100) |
| Antibody | MEF2C (Rabbit polyclonal) | Sigma-Aldrich | Cat #: HPA005533 | IF (1:100) |
| Antibody | Collagen Type VI (Mouse Monoclonal) | Sigma-Aldrich | Cat #: MAB1944 | IF (1:250) |
| Antibody | Lamin A/C (Mouse Monoclonal) | Thermo Fisher Scientific | Cat #: MA3-1000 | IF (1:100) |
| Antibody | Laminin β1 (clone 4E10) | MilliporeSigma | Cat #: MAB1921P | IF (1:100) |
| Antibody | Myosin heavy chain, neonatal | Leica Biosystems | Cat #: NCL-MHCn | IF (1:100) |
| Antibody | PAX3 | Abcam | Cat #: Ab180754 | IF (1:50) |
| Antibody | Spectrin β1 (Mouse Monoclonal) | Leica Biosystems | Cat #: NCL-SPEC1 | IF (1:50) |
| Antibody | APC Mouse Anti-Human CD56 (Mouse Monoclonal) | BD Biosciences | Cat #: 555518 | Flow (100 µl per million cells) |
| Antibody | PE anti-human CD82 (Mouse Monoclonal) | BioLegend | Cat #: 342103 | Flow (3 µl per million cells) |
| Antibody | FITC anti-human CD318 (Mouse Monoclonal) | BioLegend | Cat #: 324004 | Flow (5 µl per million cells) |
| Antibody | APC anti-human ERBB3 (Mouse Monoclonal) | BioLegend | Cat #: 324708 | Flow (5 µl per million cells) |
| Antibody | FITC anti-human NGFR (Mouse Monoclonal) | BioLegend | Cat #: 345104 | Flow (5 µl per million cells) |
| Antibody | PE anti-human CD18 (Mouse Monoclonal) | BioLegend | Cat #: 373407 | Flow (5 µl per million cells) |
| Sequence-based reagent | RT-qPCR primers | This paper | | *Supplementary file 4* |
| Commercial assay or kit | StemMACS iPS-Brew XF, human | Miltenyi Biotec | Cat #: 130-104-368 | |
| Commercial assay or kit | Skeletal Muscle Differentiation Kit | Amsbio | Amsbio Skeletal Muscle Differentiation Kit | SKM01, SKM02, and SKM03 were used for human iPSCs skeletal muscle differentiation |
| Commercial assay or kit | RNeasy Plus Mini Kit | QIAGEN | Cat #: 74136 | |
| Commercial assay or kit | Mouse on Mouse (M.O.M.) Basic Kit | Vectorlab | Cat #: BMK-2202 | |
| Commercial assay or kit | SuperScript III First-Strand Synthesis System | Invitrogen | Cat #: 18080051 | |
| Commercial assay or kit | Emerson lab custom muscle NanoString panel | NanoString Technologies | | NanoString Technologies developed the muscle NanoString panel based a gene list provided by Emerson lab |
| Commercial assay or kit | Chromium Single Cell 3' GEM, Library & Gel Bead Kit v2, 4 rxns | 10× Genomics | Cat #: 120267 | |
| Commercial assay or kit | Chromium Chip A Single Cell Kit, 16 rxns | 10× Genomics | Cat #: 1000009 | |
| Commercial assay or kit | Chromium i7 Multiplex Kit, 96 rxns | 10× Genomics | Cat #: 120262 | |
| Chemical compound, drug | ROCK Inhibitor Y-27632 | STEMCELL Technologies | Cat #: 72307 | |
| Chemical compound, drug | SB431542 | SelleckChem | Cat #: S1067 | |
| Chemical compound, drug | CHIR99021 | STEMCELL Technologies | Cat #: 72052 | |
| Chemical compound, drug | Prednisolone | Sigma-Aldrich | Cat #: P6004 | |
| Chemical compound, drug | DAPT | SelleckChem | Cat #: S2215 | |
| Chemical compound, drug | Dexamethasone | SelleckChem | Cat #: S1322 | |
| Chemical compound, drug | Forskolin | SelleckChem | Cat #: S2449 | |

*Continued on next page*

*Continued*

| Reagent type (species) or resource | Designation | Source or reference | Identifiers | Additional information |
|---|---|---|---|---|
| Software, algorithm | GraphPad Prism | GraphPad Prism, RRID:SCR_002798 | https://www.graphpad.com/ | |
| Software, algorithm | nSolver 4.0 Analysis Software | nSolver Analysis Software, RRID:SCR_003420 | http://www.nanostring.com/products/nSolver | |
| Software, algorithm | Bisulfite Sequencing DNA Methylation Analysis | BISMA, RRID:SCR_000688 | http://services.ibc.uni-stuttgart.de/BDPC/BISMA/ | |
| Software, algorithm | Cell Ranger | Cell Ranger, RRID:SCR_017344 | https://support.10xgenomics.com/single-cell-gene-expression/software/pipelines/latest/what-is-cell-ranger | Version 3.1.0 |
| Software, algorithm | STAR 2.5.1b | Implemented in Cell Ranger version 3.1.0 | https://github.com/alexdobin/STAR (*Dobin, 2022*) | Version 2.5.1b |
| Software, algorithm | Python | Python Programming Language, RRID:SCR_008394 | http://www.python.org/ | Version 2.7.9 |
| Software, algorithm | Opossum 0.2 | *Oikkonen and Lise, 2017* | https://github.com/BSGOxford/Opossum (*BSG Oxford, 2022*) | Version 0.2 |
| Software, algorithm | Platypus | Platypus, RRID:SCR_005389 | https://www.rdm.ox.ac.uk/research/lunter-group/lunter-group/all-platypus-and-stampy-versions | Version 0.8.1 |
| Software, algorithm | Demuxlet | *Kang et al., 2018* | https://github.com/statgen/demuxlet (*The Center for Statistical Genetics at the University of Michigan School of Public Health, 2021*) Version 08/03/2018 | |
| Software, algorithm | Seurat | SEURAT, RRID:SCR_007322 | http://seurat.r-forge.r-project.org/ | Version 3.1.4 |
| Software, algorithm | R Project for Statistical Computing | R Project for Statistical Computing, RRID:SCR_001905 | http://www.r-project.org/ | Version 3.6.2 |
| Software, algorithm | edgeR | edgeR, RRID:SCR_012802 | http://bioconductor.org/packages/edgeR/ | Version 3.30.3 |

## iPSC reprogramming

Human iPSCs were generated from CD56+ myoblasts or CD56− fibroblasts enriched from bicep muscle biopsies, or skin fibroblasts at the UMASS Medical School Transgenic Animal Modeling Core using CytoTune-iPS Sendai Reprogramming Kit (Thermo Fisher Scientific). Isolated iPSC clones were characterized by pluripotency identification (OCT4 staining), in vivo teratomas formation, and karyo-typing assay. iPSC lines were routinely maintained on Matrigel (Corning) with StemMACS iPS-Brew XF (Miltenyi Biotec). The cells were passaged every 4 days using StemMACS Passaging Solution XF (Miltenyi Biotec) and the Rock inhibitor Y27632 (10 µM, STEMCELL Technologies) for 24 hr to improve the survival rates. 15AM iPSCs, 15VM iPSCs, 17AM iPSCs, 17AF iPSCs, 17UM iPSCs, 30AM iPSCs, 30WM iPSCs, 54574/75 iPSCs, 54585 iPSCs, and LGMDR7 iPSCs were reprogrammed at UMass Medical School. FP and WWS iPSCs were provided by Anne Bang at Sanford Burnham Prebys Medical Discovery Institute and 54574/75 and 54585 fibroblasts were provided by Steven Moore at the University of Iowa. iPSC cell line identity was confirmed by STR profiling. iMyoblast isolation iPSCs, typically plated on six-well plates, were induced for myogenic differentiation using media prepared by Genea

Biocells and now commercially available as a skeletal muscle differentiation kit (Amsbio) following the manufacturer's specifications (*Caron et al., 2016*). After 6–7 days in S3 differentiation medium (SKM03), culture plates were rinsed with 2 ml 1× phosphate-buffered saline (PBS) and cells detached from plates in 0.5 ml TrypLE Express at 37°C and diluted with 4.5 ml HMP growth medium (Ham's F-10 supplemented with: 20% FBS, 1% chick embryo extract rich with growth factors [Emerson Lab]), 1.2 mM $CaCl_2$, 1% antibiotic/antimycotic (Gibco) (optional). The cell suspension was pipetted 5–10 times to disperse cells and clear plates of residual cells, and then 0.5–1 ml of this cell suspension was plated onto 10 cm gelatin-coated dishes in 10 ml HMP medium, cells were cultured at 5% $CO_2$ at 37°C and fed daily with fresh HMP medium to support the growth of iMyoblasts. After 2–3 culture passages, iMyoblast cells were recovered and maintained as frozen stocks for investigations. 15AM iMyoblasts, 15VM iMyoblasts, 17AM iMyoblasts, 17AF iMyoblasts, 17UM iMyoblasts, 30AM iMyo-blasts, 30WM iMyoblasts, 54574/75 iMyoblasts, 54585 iMyoblasts, LGMDR7 iMyoblasts, LGMDR9 FP iMyoblast, and WWS iMyoblasts were generated at UMass Medical School. Identity for cell lines used in single-cell RNA-sequencing was confirmed by SNP analysis. FKRP (for WWS and LGMDR9) and TCAP (for LGMDR7) mutations were confirmed by DNA sequencing and aDG expression. All cell lines have been tested for mycoplasma and have tested negative.

## Cell culture
FAC sorted CD56+ and unsorted bMyoblasts recovered from muscle biopsies (*Homma et al., 2012*; *Jones et al., 2012*) and iMyoblasts isolated from iPSC muscle cultures were routinely maintained on gelatin-coated 10 or 15 cm dishes in HMP growth medium and passaged at 70–90% confluence. To induce differentiation, cultures were grown to 95% confluence, then washed with PBS and cultured in serum-free Opti-MEM or N2 medium (DMEM/F12 supplemented with 1% N2 supplement, 1% ITS, and 1% L-glutamine) (*Barberi et al., 2007*; *Chal et al., 2016*) for 2–7 days at 37°C/5% $CO_2$. The fusion index was calculated as the percentage of nuclei in MF20-positive fibers (≥2 nuclei) to total nuclei. 15Abic biopsy myoblast, 15Vbic biopsy myoblast, 17Abic biopsy myoblast, 17Ubic biopsy myoblast, 30Abic biopsy myoblast, and 30Wbic biopsy myoblast cell lines were generated at UMass Medical School.

## Immunofluorescence
Cells were fixed on plates with 2% paraformaldehyde for 20 min at 37°C. After rinsing plates three times with PBS, the cells attached to plates were treated with blocking/permeabilizing solution (PBS containing 2% bovine albumin, 2% goat serum, 2% horse serum, and 0.2% Triton X-100) for 30 min at room temperature and then incubated with primary antibodies in PBS at 4°C overnight, washed three times in PBS, and incubated with the corresponding secondary antibodies for 1 hr at room temperature. Plates were washed two times in PBS and cells stained for 5 min with DAPI (Sigma-Aldrich) to stain nuclei and fluorescence images were taken using a Nikon Eclipse TS 100 inverted microscope.

## Flow cytometry
Single-cell suspensions of iMyoblasts and bMyoblasts cultures dissociated with TrypLE Express Enzyme (Thermo Fisher Scientific) were washed with PBS, filtered with a 40-µm strainer, and incubated with antibodies suspended in PBS for 30–60 min on ice in the dark. Cells were then washed in PBS and resuspended in PBS and 0.2% fetal calf serum, and flow cytometry was performed at UMass Medical School Flow Cytometry Core. A BD FACS Aria IIu was used for quantification and a BD FACS C-Aria II Cell Sorter was used for cell sorting. FlowJo software was used for data analysis.

## Generation of DUX4-GFP iMyoblasts
DUX4 expression was assayed in bMyoblasts and iMyoblasts expressing DUX4-GFP reporter using lentiviral vector and G418 selection (*Rickard et al., 2015*). Cells were infected using a modified spin-down method (*Springer and Blau, 1997*). In brief, $10^5$ cells per well were plated on gelatin coated six-well plates in HMP medium. The next day, cells were incubated with DUX4-GFP lentivirus in HMP medium for 15 min and then centrifuged at 1100×*g* for 30 min at 32°C. Medium containing virus was replaced with fresh medium and cells were cultured for 48 hr, then treated with 300 µg/ml G418 for 7 days for selection, with daily feedings and passaging at 90% confluence.

## Bisulfite methylation sequencing

Genomic DNA was isolated from cell pellets of FSHD1 and Ctrl ESCs (Genea Biocells) and iPSCs, S2 cells and iMyoblasts using QIAamp DNA Blood Mini Kit (QIAGEN) and bisulfite converted using the EpiTect Kit (QIAGEN) following the manufacturer's specifications. *DUX4* 4qA and 4qA-L were PCR amplified from bisulfite treated DNA using nested primers (*Jones et al., 2014*) and the *MYOD1* Core Enhancer with primers that include CpG sites 1, 2, and 3 (*Brunk et al., 1996*) as shown in *Supplementary file 1*. Amplified DNAs were cloned into the pCR2.1 TOPO vector, which was transformed into TOP10 Chemically Competent *Escherichia coli* and selected for kanamycin resistance. Cloned DNA of plasmids from kanamycin-resistant colonies were sequenced using Sanger sequencing (Sequegen), and CpG methylation analysis was analyzed by Bisulfite Sequencing DNA Methylation Analysis (BISMA) online software (*Rohde et al., 2010*).

## Single-cell RNA-seq

For each of the four cell types—cultures at stages S1, S2, iMyoblasts, and unsorted primary biopsy cells—the cells from three FSHD1 and three control subjects were detached from plates, dissociated and pooled immediately before loading ~10,000 cells on a Chromium platform (10× Genomics) for scRNA-Seq. 3′ Gene Expression v2 libraries (10× Genomics) from four Chromium runs were sequenced on an HS4K instrument at the UMMS Deep Sequencing Core. Cell Ranger version 3.1.0 (10× Genomics) and STAR 2.5.1b were used to align reads from FASTQ files to the human reference genome. Gene annotations from GRCh38.93 were prepared with cellranger mkgtf as in Cell Ranger, but with filtering modified to also retain gene biotypes 'processed_transcript' and 'bidirectional_promoter_lncRNA.' Initial filtering, barcode counting, and UMI counting yielded an estimate of 29,049 potential cell barcodes. 92.4% of the 562.5 million total reads were in cells. SNPs for each subject were genotyped from bulk RNA-seq, performed by Novogene, of S1 and iMyoblast cells for each subject in Python 2.7.9 using Opossum 0.2 (*Oikkonen and Lise, 2017*) and Platypus 0.8.1 (*Rimmer et al., 2014*). These genotypes were used to assign cells from pooled scRNA-Seq runs to their subject of origin using Demuxlet (downloaded on 08/03/2018) (*Kang et al., 2018*), and 5.7% of the cells were filtered out that did not unambiguously match a single genotype. Further filtering using the Seurat 3.1.4 package (*Butler et al., 2018*; *Stuart et al., 2019*) in R 3.6.2 removed 7.0% of the remaining cells that contained ≥12% UMIs mapped to mitochondrial genes, ≤ 1000 or ≥ 5500 detected genes, or ≥40,000 detected UMIs. This resulted in 24,991 cells, with a median of 2678 genes detected per cell and 7909 UMI per cell (*Figure 2A*). Cell clustering, cell cycle state estimation, and downstream analyses were performed for these cells in Seurat. Normalization and scaling were performed using SCTransform (*Hafemeister and Satija, 2019*) with regression against the percentage of mitochondrial gene expression. The 3000 genes with highest variability across cells were used for principal component reduction, and components 1–30 were used to construct a shared nearest neighbor graph for unsupervised cell clustering using the Louvain algorithm. This resulted in 16 cell clusters (3 for S1 cells, 4 for S2 cells, 5 for iMyoblast cells, and 4 for muscle biopsy primary cells), which were visualized using the UMAP method with resolution 0.7 (*McInnes et al., 2018*). Most of the within-cell-type subclusters were merged to reduce to six clusters, keeping separate the bMes cluster that was clearly distinct from the other three primary cell subclusters comprising bMyoblasts (*Figure 2B*), and the S2A cluster that expressed low *MYOD1* and *CDH15* and distinguished it from the other three S2 subclusters comprising more differentiated S2B cells, as described in the Results. The top 200 differentially expressed genes from each pairwise cell-type comparison using edgeR (below) were reviewed to construct a curated set of genes relevant to myogenesis or cell-type expression (shown as a Seurat dot plot in *Figure 3*).

## Differential gene expression and GO/KEGG analyses

Values from the single-cell raw count matrix (prior to normalization) were summed for each combination of the six cell types of interest (S1, S2A, S2B, iMB, bMB, and bMes) and the six donors (15A, 15V, 17A, 17U, 30A, and 30W) to obtain a table of pseudobulk counts for these 36 samples (*Tung et al., 2017*; *Supplementary file 3*). Tests of differential expression were performed on these pseudobulk counts using the R 4.0.1 package edgeR 3.30.3 (*Robinson et al., 2010*; *Lun et al., 2016*). Lowly expressed genes were filtered out with the function filterByExpr, and counts were normalized using calcNormFactors to yield values for 14,103 filtered genes across the 36 samples. We specified a quasi-likelihood

negative binomial generalized log-linear model with cell-type and donor as additive factors in the model. We used estimateDisp to estimate the dispersion for all genes, glmQLFit (robust=T) to fit a joint model to the data from all cell-types, and glmQLFTest to perform statistical tests of differential expression for each of the 15 contrasts between pairs of cell-types. The FDR was computed with the function topTags. Genes that satisfied (unadjusted) p-value<1.0E−06 and |log2(FC)|>1 were considered differentially expressed and ranked by p-value for each comparison. The results for all genes sorted by p-value are reported in **Supplementary file 1**, which also includes the FDR for each gene. Comparisons between the three FSHD1 and three control donors used the same edgeR procedure as above, but with models fit separately for each cell type, including the filterByExpr step. For GO/KEGG analyses, the differentially expressed (DE) genes, separated into up and down sets based on the sign of log2(FC), were used as DE input, and the full set of genes from the edgeR analysis after the filterByExpr step was used as the gene universe. Overrepresentation of GO terms in BP, CC, and MF ontologies (annotations from org.Hs.eg.db 3.11.4) were computed using the goana function and in KEGG pathways using the kegga function in edgeR, in both cases using log2(CPM) [CPM = counts per million] as a covariate to adjust for potential biases due to gene expression level (**Young et al., 2010**). The top GO and KEGG categories based on overrepresentation p-value, for each comparison between cell types, are summarized in **Supplementary file 2**.

## Remark on MRPL23 and AC004556.1
The gene *AC004556.1* (ENSG00000276345) is on the unlocalized chr11 scaffold KI270721.1 in GRCh38, and it differs from the chr11 gene *MRPL23* at only a single position, a common G > A variant in *MRPL23* (rs12812; allele frequency ~18%; https://www.ncbi.nlm.nih.gov/snp/rs12812). Thus, whether an RNA-seq read maps to *MRPL23* or to *AC004556.1* depends on its base-call at this variant position, and it appears that none of the Ctrl subjects has the variant but all three FSHD1 subjects are heterozygous for it. In a larger collection of FSHD1 and Ctrl subjects, this variant had allele frequency ~20% in both cases (not shown), so this is not likely to reflect an FSHD1-associated genotype, but rather a chance occurrence, and one for which the reported p-value for differential expression is artificially small because the sample variance for the number of reads assigned to *AC004556.1* has severely underestimated the population variance. Note that this G > A variant has allele frequency 17% in gnomAD 2.1.1 (https://gnomad.broadinstitute.org/variant/11-1977552-G-A?dataset=gnomad_r2_1), which uses the GRCh37 genome assembly, but is not reported at all in gnomAD 3 (https://gnomad.broadinstitute.org/variant/11-1956322-G-A?dataset=gnomad_r3), which uses the GRCh38 genome assembly. This could be due to the presence of KI270721.1 in GRCh38 but not GRCh37, and indeed the whole ~100 kb region of chr11 with high homology to KI270721.1 has low coverage in gnomAD 3 (https://gnomad.broadinstitute.org/region/11-1900000-2100000?dataset=gnomad_r3).

## Muscle xenografts
Immune deficient NOD.Cg-*Prkdc^scid^IL2r_γ^tmiWjl* /SzJ (NSG, Jackson Lab) mice that lack the ability to produce mature B cells, T cells, and natural killer (NK) cells and are highly sensitive to irradiation were used in accordance with the Institutional Animal Care and Use Committee (IACUC) at the University of Massachusetts Medical School. NSG mice were anesthetized with ketamine/xylazine and their hindlimbs were subjected to 18 Gy of irradiation using a Faxitron RV-650 or Faxitron CellRad X-ray cabinet to ablate the host mouse satellite cell population. One day after irradiation, mice were anesthetized with isoflurane and TA muscles were injected with 50 µl of 1.2% Barium Chloride (Sigma-Aldrich) bilaterally to degenerate mouse muscle. Three days after muscle injury, 1×10⁶ bMyoblasts or iMyoblasts were resuspended in 50 µl 1 mg/ml laminin (Sigma-Aldrich, L2020) in PBS and injected bilaterally into the body of TA muscles. Xenoengrafted mice were euthanized 2–4 weeks post engraftment by $CO_2$ asphyxiation followed by cervical dislocation. For IF experiments, TA muscles were isolated and embedded in optimal cutting temperature (OCT, Tissue Tek) compound, frozen in liquid nitrogen cooled isopentane and kept at –80°C until cryosectioning. For RNA isolation, xenoengrafted TA muscles were snap-frozen in liquid nitrogen and kept at –80°C until RNA extraction.

## TA sectioning and immunofluorescence
Frozen TA muscles embedded in OCT were cryosectioned using a Leica CM3050 S Cryostat. Tissue sections 10 µm thick were mounted onto Superfrost Plus glass microscope slides (Thermo Fisher

Scientific) and kept at –20°C until immunostained. When thawed, the sections were fixed with ice-cold acetone for 10 min at –20°C. We employed the 'mouse-on-mouse' (MOM) kit (Vector Laboratories) to reduce non-specific antibody staining per the manufacturer's specifications. Antibodies (key resource table) were used sequentially then slides were incubated with Hoechst block for 10 min. Following 2× for 5 min PBS washes, the slides were dried and coverslips mounted with Fluorogel. Fluorescent images were taken using a Leica DMR fluorescence microscope.

## RNA isolation, qPCR and NanoString

RNA was isolated either from cells in culture or from xenoengrafted TA muscles using the RNeasy Plus Mini Kit (QIAGEN) or Aurum Total RNA Fatty and Fibrous Tissue kit (Bio-Rad), respectively, per the manufacturer's specifications. For qPCR analysis, 2–5 µg of total RNA was converted into cDNA using the SuperScript III First-Strand Synthesis System (Invitrogen). For quantification of housekeeping genes, DUX4 target genes or muscle differentiation marker expression, 20 ng of cDNA was used for each reaction. For quantification of *DUX4* expression, 90–150 ng of cDNA was used in each reaction. For NanoString digital RNA quantification, 50 ng or 150 ng of total RNA was used for cell culture and xenografted TA muscle, respectively. An Emerson lab custom muscle NanoString panel with human-specific probes for muscle protein genes, muscle master regulatory genes, developmental transcription factors, signaling genes, and multiple housekeeping genes was used for all analyses on an nCounter Sprint profiler (NanoString Technologies, Seattle, WA). Raw mRNA counts were normalized to a panel of housekeeping genes (*RPL13A*, *GAPDH*, *GUUSB*, and *VCP*) (*Figure 7*) or just *RPL13A* (*Figure 5*) using nSolver software (NanoString Technologies, Seattle, WA). We found that normalizing to either the panel of four housekeeping genes or just to *RPL13A* gives comparable results. All qPCR and NanoString experiments included three biological replicates and each data point represents an individual biological replicate unless otherwise specified.

Statistics qPCR data are shown as the mean± SEM. Statistical differences for qPCR data were evaluated using Student's t-test and were considered significant when the p-value was less than 0.05 (*=p<0.05, **=p<0.01, ****=p<0.0001). Statistical comparisons were performed using GraphPad Prism software. Statistical methods for scRNA-Seq are described in Single-cell RNA-Seq section above.

## Acknowledgements

The authors thank the Emerson lab for thoughtful discussions, Xiaoling Chen for contributions to WWS Western blot assays, Teagan Parsons for scRNA-Seq technical advice, Dr. Ellie Kittler for RNA-seq sequencing, Dr. Steven Moore, University of Iowa, for infantile FSHD1 fibroblasts, Dr. Jenny Morgan for advice on human-specific muscle antibodies, Dr. Jiri Vajsar at the Hospital for Sick Children, Toronto for providing Walker Warburg (WWS) patient cells and Lila Habib at Sanford Burnham Prebys Medical Discovery Institute for contributing to WWS iPSC generation. The authors thank Peter and Takako Jones for advice about bisulfite DNA sequencing. Funded by grants to CPE from the Muscular Dystrophy Association (480265), LGMD 2i Fund, NICHD Wellstone Muscular Dystrophy Cooperative Research Center P50 HD060848, and NICHD 3R37HD007796, and to LJH from the Muscular Dystrophy Association (577797) and the Wellstone Center.

## Additional information

### Competing interests

Dongsheng Guo: Co inventor on "Microhomology Mediated Repair Of Microduplication Gene Mutations" (17/051,632) and "Methods And Compositions For Treatment Of Muscle Disease With iPSC-Induced Human Skeletal Muscle Stem Cells". Katelyn Daman, Meng-Jiao Shi, Jing Yan: Co-inventor on "Methods And Compositions For Treatment Of Muscle Disease With iPSC-Induced Human Skeletal Muscle Stem Cells". Jennifer JC Chen: Co-inventor on "Methods And Compositions For Treatment Of Muscle Disease With iPSC-Induced Human Skeletal Muscle Stem Cells" and "Microhomology Mediated Repair Of Microduplication Gene Mutations" (17/051,632). Amanda M Rickard, Monica H Bennett: Was affiliated with Genea BioCells. This author has no financial interests to declare. Alex Kiselyov: Was a formerly affiliated with Genea BioCells. The author has no financial

interests to declare. Oliver D King: Co-inventor on "Molecular diagnosis of FSHD by epigenetic signature" (US10870886B2) and "Microhomology Mediated Repair Of Microduplication Gene Mutations" (17/051,632) and "Methods And Compositions For Treatment Of Muscle Disease With iPSC-Induced Human Skeletal Muscle Stem Cells". Lawrence J Hayward: Co-inventor on "Methods And Compositions For Treatment Of Muscle Disease with iPSC-Induced Human Skeletal Muscle Stem Cells". Charles P Emerson: Co-inventor on "Microhomology Mediated Repair Of Microduplication Gene Mutations" (17/051,632) and "Methods And Compositions For Treatment Of Muscle Disease With iPSC-Induced Human Skeletal Muscle Stem Cells". The other authors declare that no competing interests exist.

## Funding

| Funder | Grant reference number | Author |
| --- | --- | --- |
| Muscular Dystrophy Association | 480265 | Charles P Emerson |
| National Institutes of Health | HD060848 | Charles P Emerson Lawrence Hayward |
| National Institutes of Health | 3R37HD007796 | Charles P Emerson |
| Muscular Dystrophy Association | 577797 | Lawrence Hayward |
| LGMD 2I Fund | | Charles P Emerson |

The funders had no role in study design, data collection and interpretation, or the decision to submit the work for publication.

## Author contributions

Dongsheng Guo, Katelyn Daman, Data curation, Formal analysis, Investigation, Writing – original draft, Writing – review and editing; Jennifer JC Chen, Conceptualization, Data curation, Formal analysis, Investigation, Writing – review and editing; Meng-Jiao Shi, Jing Yan, Zdenka Matijasevic, Oliver D King, Data curation, Formal analysis, Investigation, Writing – review and editing; Amanda M Rickard, Monica H Bennett, Alex Kiselyov, Data curation, Writing – review and editing; Haowen Zhou, Anne G Bang, Data curation, Investigation, Writing – review and editing; Kathryn R Wagner, René Maehr, Resources, Writing – review and editing; Lawrence J Hayward, Data curation, Formal analysis, Writing – original draft, Writing – review and editing; Charles P Emerson, Conceptualization, Formal analysis, Funding acquisition, Resources, Supervision, Writing – original draft, Writing – review and editing

## Author ORCIDs

Dongsheng Guo http://orcid.org/0000-0002-9702-7327
Katelyn Daman http://orcid.org/0000-0003-1088-3903
René Maehr http://orcid.org/0000-0002-9520-3382
Oliver D King http://orcid.org/0000-0002-9460-4491
Lawrence J Hayward http://orcid.org/0000-0003-0579-0358
Charles P Emerson Jr, http://orcid.org/0000-0003-3744-9090

## Ethics

Human subjects: Informed consent was obtained from patients who donated tissue for production of cell lines used in these studies. IRB protocols approved by UMass Medical School IRB: H00006581-10 and H00006581-11; IRB protocol approved by Kennedy Krieger Institute IRB: B0410080117; IRB protocol approved by the University of Iowa IRB: 200510769.

Animals were used in accordance with the Institutional Animal Care and Use Committee (IACUC) at the University of Massachusetts Medical School (protocol number PROTO201900322).

## Decision letter and Author response

Decision letter https://doi.org/10.7554/eLife.70341.sa1
Author response https://doi.org/10.7554/eLife.70341.sa2

## Additional files

### Supplementary files

• Supplementary file 1. Differential expression for pairwise cell type comparisons from edgeR analysis. For each of the 15 pairwise comparisons (shown in separate tabs), genes with differential expression (in either up or down directions) were ranked by p-value if the *P*-value < 1.0E-06 and |log2(FC)| > 1. The table includes columns for p-values, log2(FC), log2(CPM), QL F-test (F), and false discovery rate (FDR) from edgeR, and several columns for gene annotations.

• Supplementary file 2. Top 20 Up and Down GO terms (BP, CC, MF) and KEGG pathways for pairwise cell type comparisons. For each of the 15 pairwise comparisons (shown in separate tabs), the goana and kegga functions in edgeR were used to rank the top 20 Gene Ontology (GO) terms from each of the biological process (BP), cellular component (CC) and molecular function (MF) branches of GO, and the top 20 KEGG pathways, based on overrepresentation among the DE genes. Terms comprising N ( < 500) genes are sorted by P.Up and P.Down. For each term, the top DE Genes.Up and Genes.Down (ordered by P Value) are listed, up to a maximum of 30 genes.

• Supplementary file 3. Pseudobulk counts for 36 samples based on scRNA-Seq data. Raw single-cell counts were summed for each of the 6 cell types or subclusters of interest (S1, S2A, S2B, iMyoblast, bMyo, bMes) in each of the six donors (15 A, 15 V, 17 A, 17 U, 30 A, 30 W).

• Supplementary file 4. Primer sequences. Table lists all primers for qPCR and bisulfite sequencing.

• Transparent reporting form

### Data availability

Due to IRB restrictions and privacy considerations, the human high-throughput sequencing data (RNA-seq and scRNA-seq) generated in this study cannot be made accessible in a public repository, but is being deposited in dbGaP (accession phs002438.v1.p1) for controlled access by qualified researchers (from not-for-profit organizations, for FSHD-related research, with local IRB approval). We do however provide processed versions of this data, deposited at Zenodo (https://zenodo.org/record/4839099): an R Seurat object that includes read counts for each gene in each cell, and a table of pseudobulk counts for each gene in each sample (also in Supplementary file 3), along with the R code for the figures and analyses that use this processed data (Figures 2 and 3; Supplementary files 1 and 2).

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
