## [Editor Report]

This is an interesting and systematically constructed paper developing an iPSC–myoblast platform. It covers the generation of the cell system, and its detailed description. Assessments are made as to the extent this iPSC system recapitulates Limb Girdle Muscular Dystrophy and Walker Warburg Syndrome. The findings promote the view that iPSC myoblasts have a potential in studying muscle stem cell functions and for therapeutic development.

---

## [Decision Letter]

**Decision letter after peer review:Decision letter**

[Editors’ note: the authors submitted for reconsideration following the decision after peer review. What follows is the decision letter after the first round of review.]

Thank you for submitting the paper "iMyoblasts for ex vivo and in vivo investigations of human myogenesis and disease modeling" for consideration by *eLife*. Your article has been reviewed by 2 peer reviewers, one of whom is a member of our Board of Reviewing Editors, and the evaluation has been overseen by a Senior Editor. The reviewers have opted to remain anonymous.

We are sorry to say that, after consultation with the reviewers, we have decided that this work will not be considered further for publication by *eLife*. However, whilst this is a reject decision owing to the substantial issues raised in particular by the second reviewer, this reflects the large number of issues that we consider will require significant time to address. Both the editors and the reviewers are sympathetic to the paper and would be open to seeing a new revised submission.*Reviewer #1:*

This is an interesting and systematically constructed paper developing an iPSC–myoblast platform. It covers the generation of the cell system, and its detailed description. Assessments are made as to the extent this iPSC system recapitulates Limb Girdle Muscular Dystrophy and Walker Warburg Syndrome. The findings promote the view that iPSC myoblasts have a potential in studying muscle stem cell functions and for therapeutic development.

This is a well written and thorough study. Subject to the expert advice of remaining reviewers more directly within this field, the authors have demonstrated that in:

1) A capacity for the iPSC Myoblasts to be generated and their commitment to form viable primary lines committed to differentiate in Opti–MEM serum free medium [as reflected in qPCR assays].

2) FSHD1 and Ctrl iPSCs and ESCs, and LGMD R7 and WWS iPSCs, qPCR demonstrated parallel changes in expression of the MYOD1 and PAX3 regulatory and MYH8, myosin heavy chain genes.

3) The iPSC myoblasts constituting a specific lineage, reflected in their transcriptosomes and differentiation and xenograft properties.

The authors thus have developed a successful and efficient and reliable method to isolate iPSC myoblasts. This is an attractive and thorough study.

This reviewer would suggest a survey of future studies that may be performed of the functional, including the contractile and electrophysiological properties of this cell system, including its excitation contraction coupling properties.

*Reviewer #2:*

The manuscript by Guo et al. describes a method to select an expandable population of myogenic reserve cells from a commercially available protocol to differentiate human pluripotent cells into skeletal myogenic cells. The study is potentially interesting for the rapidly growing community of muscle researchers utilizing iPSCs for disease modelling and in vivo studies, but it is unclear what the key advantage over other methods and over the commercial kit is. The expectation from such a method would have been to have more mature, adult–like myogenic progenitors, as most small molecule based hiPSC-differentiation methodologies give rise to embryonic/fetal myoblasts. However, this appears not to be the case, as Pax7 remains low and Figure 3 shows that NFIX is only expressed in the biopsy–derived population. The authors mention that iMyoblasts are a distinct lineage, but do not provide additional information on whether this is a culture artefact or a real developmental lineage captured in vitro.

The downstream applications of this protocol shown in this study are not novel and overall, difficult to follow. it is unclear if the authors wanted to validate a new methodology using established readouts or show that their methodology can unravel new findings. Overall, having three different disease targets, in vito and in vivo models makes the story somehow difficult to follow.

Please find below a list of some general issues:

– Genea company and differentiation kit no longer exist and no mention about validation of this work with the new Myocea kit.

– Pax 3 and MyoD not quantified at the protein level.

– Fusion stated to be a key difference between iMyoblasts and S3 cells, but not quantified.

– Tertiary reserve population experiment not quantified.

– In Material and Methods it is said that a differentiation of 2 to 7 days was performed. This sounds like a rather wide spectrum. Cells differentiated for 3 days could show a profile quite different from cells that have been kept in differentiation for 7 days.

In vivo section:

– No quantification of LAMIN A/C & SPECTRIN double positive donor derived fibers provided.

– Given the capacity of forming reserve cells, the expectation in this study was to see this finding validated also in vivo with serial injuries or transplantation, as well as quantification of contribution to satellite cell pool.

Disease modelling:

– The section on FSHD is perhaps overly technical and difficult to digest for non specialists. It was complicated to grasp what the actual advance is in terms of in vitro FSHD modelling in this study, particularly in view of the fact that this manuscript focus on FSHD as main disease model, which is also the focus of the original Caron et al.,/Genea paper describing this methodology.

– FKRP: in this case the novelty is limited (e.g. https://pubmed.ncbi.nlm.nih.gov/31566294/) and the reported findings are very limited in terms of disease modelling.

– LGMDR7: this section appears out of context and brief.

Figure 1:

– Panel C is not discussed/touched in Results section. What is the message of that panel in the context of figure 1? (see general comments).

– In figure legend: unclear what Genea 002, 015, 019, etc. mean. Please specify this in Material and Methods as well as Results sections.

Figure 2: lack of IF quantifications (MyoD, pax3, pax7, fusion index…).

Figure 3: conclusion/take home message unclear.

Figure 4: Dux4 up regulation from reporter line not quantified.

Significant work is required to reshape the form and message of this manuscript, including adding more data.

Please find below a list of some general issues:

– Genea company and differentiation kit no longer exist and no mention about validation of this work with the new Myocea kit – Please validate key findings of this study with Myocea kit.

– Pax 3 and MyoD not quantified at the protein level – please quantify.

– Fusion stated to be a key difference between iMyoblasts and S3 cells, but not quantified – please quantify.

– Tertiary reserve population experiment not quantified – please quantify.

In vivo section:

– No quantification of LAMIN A/C & SPECTRIN double positive donor derived fibers provided. – please quantify.

– Given the capacity of forming reserve cells, the expectation in this study was to see this finding validated also in vivo with serial injuries or transplantation, as well as quantification of contribution to satellite cell pool – please quantify donor derived Pax7+ cells in vivo and perform serial injury or transplantation study and quantification.

Disease modelling:

– FSHD– please clarify take home message and quantify Dux4 from reporter cell line.

---

## [Author Response]

[Editors’ note: The authors appealed the original decision. What follows is the authors’ response to the first round of review.]

Reviewer #1:This is an interesting and systematically constructed paper developing an iPSC–myoblast platform. It covers the generation of the cell system, and its detailed description. Assessments are made as to the extent this iPSC system recapitulates Limb Girdle Muscular Dystrophy and Walker Warburg Syndrome. The findings promote the view that iPSC myoblasts have a potential in studying muscle stem cell functions and for therapeutic development.

Reviewer #1 recognizes the novel applications of our iMyoblast technology for studies of human muscle stem cell biology and therapeutic development for multiple muscular dystrophies. We also note that our findings open exciting opportunities for development of muscle stem cell therapeutics based on our discovery that iMyoblasts can be expanded to large numbers in cell culture (Figure 1), iMyoblast xenograft muscle fibers show regulatory plasticity to undergo adult Myosin isoform switching following engraftment into fast adult TA muscle of the mouse (Figure 8), and new data showing that iMyoblast xenografts produce differentiated muscle as well as a population of undifferentiated iMyoblasts that can regenerate xenograft muscle in vivo in response to secondary injury (Figure 10).

This is a well written and thorough study. Subject to the expert advice of remaining reviewers more directly within this field, the authors have demonstrated that in:1) A capacity for the iPSC Myoblasts to be generated and their commitment to form viable primary lines committed to differentiate in Opti–MEM serum free medium [as reflected in qPCR assays].

In addition to Opti-MEM, our qPCR and NanoString RNA expression studies show iMyoblasts differentiate robustly in N2 serum free medium as well as other published specialized myoblast differentiation media (Figures 4, 5 and Figure 4—figure supplement 1). Notably, these data show that the gene expression response of iMyoblasts to differentiation media is distinct from that of S2 stage embryonic myogenic progenitors and adult biopsy bMyoblasts.

2) FSHD1 and Ctrl iPSCs and ESCs, and LGMD R7 and WWS iPSCs, qPCR demonstrated parallel changes in expression of the MYOD1 and PAX3 regulatory and MYH8, myosin heavy chain genes.3) The iPSC myoblasts constituting a specific lineage, reflected in their transcriptosomes and differentiation and xenograft properties.The authors thus have developed a successful and efficient and reliable method to isolate iPSC myoblasts. This is an attractive and thorough study.This reviewer would suggest a survey of future studies that may be performed of the functional, including the contractile and electrophysiological properties of this cell system, including its excitation contraction coupling properties.

As suggested by Reviewer 1, our revised manuscript discusses future studies that can be uniquely performed with iMyoblasts and are ongoing in our lab. These include:

– Contraction and electrophysiological assays using 3D iMyotube cultures to model muscle fiber maturation, stem cell niche selection, muscle disease pathology, and validate therapeutics;

– iMyoblast gene editing investigations to correct genes causing muscular dystrophies;

– Gene editing investigations of the muscle stem cell functions of iMyoblast genes identified in scRNA-Seq transcriptome studies;

– Epigenetic mechanisms regulating the FSHD disease gene, DUX4, as discovered in this study;

– iMyoblast stem cell therapeutics exploiting the capacity of iMyoblasts for expansion to large cell numbers, muscle gene regulatory plasticity to respond to the adult muscle environment in xenografts, and iMyoblast muscle regeneration in response to injury in vivo.

Reviewer #2:The manuscript by Guo et al. describes a method to select an expandable population of myogenic reserve cells from a commercially available protocol to differentiate human pluripotent cells into skeletal myogenic cells. The study is potentially interesting for the rapidly growing community of muscle researchers utilizing iPSCs for disease modelling and in vivo studies, but it is unclear what the key advantage over other methods and over the commercial kit is. The expectation from such a method would have been to have more mature, adult–like myogenic progenitors, as most small molecule based hiPSC-differentiation methodologies give rise to embryonic/fetal myoblasts. However, this appears not to be the case, as Pax7 remains low and Figure 3 shows that NFIX is only expressed in the biopsy–derived population. The authors mention that iMyoblasts are a distinct lineage, but do not provide additional information on whether this is a culture artefact or a real developmental lineage captured in vitro.

Our iMyoblast technology is unique. This technology employs a commercially available (AMSBIO) gene free myogenic induction, similar to other iPSC induction protocols, but this induction protocol is used in combination with a reserve cell selection protocol not used by other protocols. Reserve cell selection recover iMyoblast with stem cell capacity to become quiescent in differentiated muscle cultures analogous to satellite cells. iMyoblasts retain dual potential for regeneration and differentiation both ex vivo (Figure 1—figure supplement 2) and in vivo in response to injury (Figure 10). Furthermore, our reserve cell protocol reproducibly generates a homogeneous, expandable population of PAX3/MYOD1 iMyoblasts. Expansion has not yet been demonstrated by other iPSC gene free myogenic induction protocols, which require FAC sorting to recover minor cell populations which also differ from iMyoblasts in their expression of PAX7. The capacity of iMyoblasts for expansion enables statistically powered ex vivo and in vivo molecular, cellular and stem cell studies of human myogenesis and disease pathology, as we demonstrate in our study.

– scRNA-Seq studies show that iMyoblasts have a transcriptome that is distinct from both S1 and S2 myogenic progenitors produced by the AMSBIO commercial iPSC induction kit used in the first step of our iMyoblast isolation protocol and from adult bMyoblasts (Figure 2). iMyoblasts fuse and activate FSHD disease genes to high levels similar to adult biopsy bMyoblasts, whereas S1 and S2 cells differ in their differentiation as mononucleated iMyocytes typical of the first formed embryonic muscles (Figures 4, 5 and Figure 4—figure supplement 2) and activate FSHD disease genes at very low levels (Figure 5C), further confirming that iMyoblasts are a different myogenic cell. In revision, we discuss evidence that iMyoblast transcriptome also is distinct from the transcriptomes of iPSC-induced *PAX7* Myoblasts generated by published protocols, although both PAX3 iMyoblasts and iPSC induced PAX7 cells share the fetal-like gene expression profiles along the developmental trajectory of limb myogenic cells as described by Xi et al. In light of of these considerations, in the revised manuscript, we no longer refer to iMyoblasts as a “distinct lineage”.

– Regarding the reviewer’s expectation of seeing markers of more mature/adult-like myogenic progenitors: iMyoblasts express CD82, an established marker of fetal myoblasts (Figure 1D), and the differentiation gene expression profiles of iMyoblasts ex vivo in culture are embryonic/fetal as shown by their expression of embryonic MYH8 in response to iMyotube differentiation (Figure 4). Notably, this embryonic/fetal ex vivo signature can switch in vivo in iMyoblast xenograft muscle, which upregulate the expression of adult Myosin isoforms and down regulate embryonic/fetal isoforms following engraftment, revealing the developmental plasticity and capacity of iMyoblast xenografts to transcriptionally switch towards adult muscle phenotypes in response to the in vivo environment of adult muscle (Figures 7 and 8).

– In the case of NFIX expression, we note in the revision that our scRNA-Seq data (Figure 2C and Supplementary file 3) show that iMyoblasts express NFIX although at somewhat lower levels than biopsy myoblasts, consistent with their fetal-like phenotype, but at higher levels than S1 and S2 stage myogenic cells, consistent with their embryonic-like iMyocyte phenotype. We also note that NFIX is not a myogenic-specific gene based on its abundant expression by bMes non-myogenic cells isolated from muscle biopsies (Figure 2C).

– iMyoblasts are a PAX3 myogenic progenitor cells. In mouse, different neonatal and adult muscles are populated with both PAX3 and PAX7/PAX3 myogenic cells (PMID: 31006622; PMID: 31006621). In humans, we know little about PAX3 and PAX7 myogenic progenitor populations and these may be divergent from the mouse, as suggested by recent human genetic evidence that PAX7 is not an essential human muscle gene as it is in the mouse (PMID: 31852888; PMID: 31092906). Therefore, we do not draw lineage inferences about PAX3 iMyoblasts.

The revised manuscript addresses and clarifies all of the above issues.

The downstream applications of this protocol shown in this study are not novel and overall, difficult to follow. it is unclear if the authors wanted to validate a new methodology using established readouts or show that their methodology can unravel new findings. Overall, having three different disease targets, in vito and in vivo models makes the story somehow difficult to follow.

We strongly disagree with Reviewer 2’s assertion that the downstream applications of this protocol are not novel. While some of the experiments may be viewed as validation that iMyoblasts behave similarly to primary biopsy bMyoblasts in certain regards, iMyoblasts enable applications that would not be possible or practical with bMyoblasts.

– Expandable populations of human iMyoblasts from patients with genetic diseases (Figure 1C) enable statistically powered and reproducible qPCR and NanoString gene expression investigations of human myogenesis and disease pathology, as illustrated in our cell culture studies (Figures 4, 5, & Figure 4—figure supplement 1) and in vivo in iMyoblast-generated xenograft studies (Figures 7, 9 & 10). As such, iMyoblasts capacity for robust myotube differentiation also will enable proteomic and electrophysiological studies of patient and control muscle in 3D culture models;

– Expandable populations of human iPSCs from patients enable development of human autologous transplantation technologies to repair disease and damaged muscles, a technology that in the past has been limited by the inability to produce large numbers of myoblasts because of patient age or disease progression;

– CRISPR gene editing is efficient in iMyoblasts compared to adult bMyoblasts, as we have reported in studies of TCAP iMyoblasts (Iyer et al., Nature, 2019), and in combination with the capacity of iMyoblasts for expansion, iMyoblasts uniquely enable development of disease gene editing strategies for multiple genetic muscle diseases. Additionally, CRISPR mutagenesis and gene activation/repression technologies are being used to investigate the functions of specific muscle regulatory genes (identified in our scRNA-Seq studies) in the regulation of iMyoblast identity, stem cell behavior and the initiation of differentiation, processes that can be investigated both ex vivo and in vivo in iMyoblast muscle xenografts;

– iMyoblasts can be readily produced from patient iPSC cells for drug screening studies, as shown by our finding that FSHD iMyoblasts are responsive to established FSHD drugs ex vivo (Figure 5—figure supplement 1), and support future delivery and target engagement studies in vivo in xenograft muscles (Figure 7).

– iMyoblasts efficiently engraft and undergo adult MYH isoform switching in response the in vivo muscle environment, uniquely enabling in vivo experimental investigations of MYH isoform switching, innervation, and vascularization (Figures 7B, Figure 7—figure supplement 1 & 8).

– iMyoblasts isolated as reserve cells behave as an expandable regenerative stem cell, both ex vivo (Figure 1 figure supplement 2) and in vivo (Figure 10). This enables studies to use iMyoblasts as a stem cell source for development of muscle stem cell therapeutics as well as a model for gene editing studies to identify genes that control stem cell behavior and to modify iMyoblast gene expression to enhance in vivo delivery and engraftment.

Please find below a list of some general issues:– Genea company and differentiation kit no longer exist and no mention about validation of this work with the new Myocea kit.

All of our culture and iMyoblast isolation studies were performed using the Genea/Myocea VC media components, which are commercially available from AMSBIO (Skeletal Muscle Differentiation kit), as confirmed by Myocea. Our revised manuscript clarifies primary induction media sources and their current commercial availability.

– Pax 3 and MyoD not quantified at the protein level.

The reviewer’s rationale for requesting PAX3 and MYOD protein expression data is unclear and we believe unnecessary given our mRNA expression data and previous results from Caron et al. In response to Reviewer 2’s request, the revised manuscript includes immunofluorescence data and quantitation of MYOD1 (Figure 1A) and PAX3 (Figure 10) expression.

– Fusion stated to be a key difference between iMyoblasts and S3 cells, but not quantified.

The revised manuscript provides S3 iMyocyte and iMyoblast fusion index data (Figure 1A) showing that S3 iMyocytes differentiate as a largely mononucleated cell population typical of embryonic muscle whereas iMyoblasts fuse to form myotubes as is typical of fetal, neonatal and adult myoblasts. In addition to these differences in fusion:

– iMyoblasts and S1 and S2 stage cells have distinct transcriptomes (Figure 2);

– iMyoblasts activate DUX4 biomarkers with similar kinetics and levels to biopsy bMyoblasts (Figure 5 & Figure 4—figure supplement 1), unlike S3 i which are epigenetically repressed for DUX4 activation (Figure 6A and Figure 6—figure supplement 1C);

– iMyoblasts respond differentially to a panel of specialized differentiation media than S3 cells, as assayed by expression of myogenic regulatory RNAs and muscle protein RNAs (Figure 4B).

– Tertiary reserve population experiment not quantified.

The revised manuscript includes quantitative IF data for the tertiary reserve population (Figure 1—figure supplement 2).

– In Material and Methods it is said that a differentiation of 2 to 7 days was performed. This sounds like a rather wide spectrum. Cells differentiated for 3 days could show a profile quite different from cells that have been kept in differentiation for 7 days.

The revised manuscript clarifies the Material and Methods to state that iMyoblast and biopsy bMyoblast differentiation was quantitated by kinetic assays of gene expression at multiple time points (2, 4 and 7 days) following addition of differentiation medium (Figure 4, Figure 4—figure supplement 1 & Figure 5).

In vivo section:– no quantification of LAMIN A/C & SPECTRIN double positive donor derived fibers provided.

In our revised manuscript, quantification of LAMIN A/C & SPECTRIN double positive fibers is shown in (Figure 7B). In addition, the revised manuscript includes NanoString data comparing expression of human to mouse housekeeping and muscle genes in muscle xenografts, an approach that provides quantitative measures of engraftment efficiency (Figure 7D).

– Given the capacity of forming reserve cells, the expectation in this study was to see this finding validated also in vivo with serial injuries or transplantation, as well as quantification of contribution to satellite cell pool.

The revised manuscript includes new results of our secondary injury studies, supporting that iMyoblasts contribute to a satellite cell-like pool of regenerative cells as well as a source of stem cells for muscle fiber differentiation (Figure 10). Quantification of PAX3 immunostaining is shown for primary and secondary injury xenografts.

Disease modelling:– the section on FSHD is perhaps overly technical and difficult to digest for non specialists. It was complicated to grasp what the actual advance is in terms of in vitro FSHD modelling in this study, particularly in view of the fact that this manuscript focus on FSHD as main disease model, which is also the focus of the original Caron et al.,/Genea paper describing this methodology.

FSHD is a complex epigenetic disease, and the revised manuscript strives to improve the presentation of FSHD iMyoblast data in the context of iMyoblasts. Our FSHD iMyoblast findings highlight the power of iMyoblast and iPSC technology to investigate the epigenetic as well as genetic mechanisms that regulate the DUX4 disease gene through a DNA methylation repression mechanism. Our FSHD findings challenge the findings of Caron et al., 2016 by showing that DUX4 expression is highly repressed in S3 myocytes during primary induction of FSHD patient iPSCs using the same FSHD patient ESCs initially reported on by Caron et al. (Figures 6A and Figure 1—figure supplement 1). Our findings also show that this DUX4 repression is mediated by a DNA methylation mechanism (Figure 6E), and that FSHD iMyoblasts have similar kinetics of upregulation and levels of DUX4 expression during differentiation as we show for FSHD adult biopsy Myoblasts (Figure 5A), establishing their utility for ex vivo and in vivo investigations of FSHD molecular pathology.

– FKRP: in this case the novelty is limited (e.g. https://pubmed.ncbi.nlm.nih.gov/31566294/) and the reported findings are very limited in terms of disease modelling.

Investigations of FKRP iMyoblasts validate the utility of iMyoblasts for investigating the muscle pathology in FKRP dystrophies. Reviewer 2 cites a paper that uses iPSCs to model the neurological impact of FKRP dystrophies, by differentiating them into cortical neurons, not muscle. Notably, our findings show that iMyoblasts can model both the clinically severe FKRP disease alleles (Walker Warburg) that causes both neurological and muscle pathology as well as the more common LGMDR9 alleles, that causes only muscle disease, based on levels of aDG glycosylation (Figure 9F). We are currently using FKRP iMyoblasts to develop a novel universal FKRP gene editing strategy for correcting FKRP disease alleles, work that will be published separately.

– LGMDR7: this section appears out of context and brief.

Investigations of LGMDR7 iMyoblasts further validate the utility of iMyoblasts for investigating the molecular pathology of muscular dystrophies in general and highlight specifically the utility of LGMDR7 iMyoblasts for studies using disease gene editing, as we have reported (Iyer et al. Nature, 2019).

The revised manuscript emphasizes the specific contributions of iMyoblasts to model human muscular dystrophies.

Figure 1:– Panel C is not discussed/touched in Results section. What is the message of that panel in the context of figure 1? (see general comments).

In the revised manuscript, Figure 1, Panel C is introduced later in the section on FSHD iMyoblast modeling, which we agree was awkwardly presented in the original manuscript. This and other figures have been reorganized in the revised manuscript to more effectively integrate iMyoblasts characterization and disease modeling in the revised manuscript.

– In figure legend: unclear what Genea 002, 015, 019, etc. mean. Please specify this in Material and Methods as well as Results sections.

Genea 002, 015 and 019 refer to the ESC cell line designation reported in Caron et al. paper, and we clarified this in the revised manuscript. These FSHD ESC lines were utilized to validate our unexpected finding of DUX4 repression in FSHD patient iPSC during primary induction of myogenesis and exclude the possibility that this finding was an artifact of iPSC reprogramming. We show that Genea/Myocea FSHD ESCs and our iPSCs do not activate DUX4, which remains repressed by a DNA methylation mechanism, challenging the conclusions of the Caron et al. paper on DUX4 expression.

Figure 2: lack of IF quantifications (MyoD, pax3, pax7, fusion index…).

As discussed above, the revised manuscript provides IF quantification for MYOD1 and fusion indices in Figure 1A and Figure 1—figure supplement 2.

Figure 3: conclusion/take home message unclear.

The revised manuscript clarifies the impact of scRNA-Seq UMAP and edgeR analyses for identification of iMyoblasts as a distinct myogenic cell, including further discussion of NFIX target genes as markers of a fetal cell lineage, as elaborated above.

Figure 4: Dux4 up regulation from reporter line not quantified.

The revised paper provides quantitative data on reporter gene expression (Figure 5B), to complement the quantitative qPCR data showing that FSHD patient iMyoblasts and biopsy bMyoblasts upregulate DUX4 to similar levels in response to differentiation.

Significant work is required to reshape the form and message of this manuscript, including adding more data.Please find below a list of some general issues:– Genea company and differentiation kit no longer exist and no mention about validation of this work with the new Myocea kit – Please validate key findings of this study with Myocea kit.– Pax 3 and MyoD not quantified at the protein level – please quantify.– Fusion stated to be a key difference between iMyoblasts and S3 cells, but not quantified – please quantify.– Tertiary reserve population experiment not quantified – please quantify.

These concerns have been fully addressed in revision, as discussed in detail above.

In vivo section:– no quantification of LAMIN A/C & SPECTRIN double positive donor derived fibers provided. – please quantify.– given the capacity of forming reserve cells, the expectation in this study was to see this finding validated also in vivo with serial injuries or transplantation, as well as quantification of contribution to satellite cell pool – please quantify donor derived Pax7+ cells in vivo and perform serial injury or transplantation study and quantification.

These concerns have been fully addressed in revision, as discussed in detail above.

Disease modelling:– FSHD– please clarify take home message and quantify Dux4 from reporter cell line.

These concerns have been fully addressed in revision, as discussed in detail above.